



# 1 First Atmospheric Aerosol Monitoring Results from Geostationary
# 2 Environment Monitoring Spectrometer (GEMS) over Asia

Yeseul Cho[1], Jhoon Kim[1], Sujung Go[2,3], Mijin Kim[4], Seoyoung Lee[2,3], Minseok Kim[1], Heesung Chong[5],
Won-Jin Lee[6], Dong-Won Lee[6], Omar Torres[3], Sang Seo Park[7]
[1]Department of Atmospheric Sciences, Yonsei University, Seoul, Republic of Korea
[2]Goddard Earth Sciences Technology and Research (GESTAR) II, University of Maryland, Baltimore County, Baltimore, MD
21250, USA
[3]NASA Goddard Space Flight Center, Greenbelt, MD, USA
[4]Goddard Earth Sciences Technology and Research (GESTAR) II, Morgan state university, Baltimore, MD 21251, USA
[5]Center for Astrophysics | Harvard & Smithsonian, Cambridge, MA 02138, USA
[6]National Institute of Environmental Research, Incheon, Republic of Korea
[7]Department of Civil, Urban, Earth and Environmental Engineering, Ulsan National Institute of Science and Technology,
Ulsan, Republic of Korea
*Correspondence to*: Jhoon Kim (jkim2@yonsei.ac.kr)
**Abstract.** Aerosol optical properties have been provided from the Geostationary Environment Monitoring Spectrometer
(GEMS). It is the world's first geostationary earth orbit (GEO) satellite instrument designed for atmospheric environmental
monitoring. This study describes improvements to the GEMS aerosol retrieval algorithm (AERAOD). These include spectral
binning, surface reflectance estimation, cloud masking, and post-processing. Furthermore, the study presents validation results.
These enhancements are aimed at providing more accurate and reliable aerosol monitoring results for Asia. The adoption of
spectral binning in the lookup table (LUT) approach reduces random errors and enhances the stability of the satellite
measurements. In addition, we introduce a new high-resolution database for surface reflectance estimation based on the
minimum reflectance method adapted to the GEMS pixel resolution. Monthly background aerosol optical depth (BAOD)
values are used to consistently estimate the hourly GEMS surface reflectance. Advanced cloud-removal techniques are
implemented to significantly improve the effectiveness of cloud detection and enhance the quality of aerosol retrieval. An
innovative post-processing correction method based on machine learning is introduced to address artificial diurnal biases in
aerosol optical depth (AOD) observations. This study investigates specific aerosol events. It highlights capability of GEMS to
monitor and provide insights into hourly aerosol optical properties during various atmospheric events. The performance of the
GEMS AERAOD products is validated against the Aerosol Robotic Network (AERONET) and Cloud-Aerosol Lidar with
Orthogonal Polarization (CALIOP) data for the period from November 2021 to October 2022. The GEMS AOD demonstrates
a strong correlation with the AERONET AOD (R = 0.792). However, it exhibits bias patterns including underestimation of
high AOD values and overestimation in low AOD conditions. Different aerosol types (highly absorbing fine, dust, and non-
absorbing) exhibit distinct validation results. The GEMS single scattering albedo (SSA) retrievals agree well with the
AERONET data within reasonable error ranges, with variations observed among the aerosol types. For GEMS AOD exceeding
0.4 (1.0), 42.76% (56.61%) and 67.25% (85.70%) of GEMS SSA data points fall within the ±0.03 and ±0.05 error bounds,
respectively. Model-enforced post-processing correction improved the GEMS AOD and SSA performances, thereby reducing
the diurnal variation in biases. The validation of the GEMS aerosol layer height (ALH) retrievals against the CALIOP data
demonstrates a good agreement, with a mean bias of -0.225 km, and 55.29% (71.70%) of data within ±1 km (1.5 km).



## 1 Introduction

The regional and global monitoring of aerosol optical properties (AOPs) was conducted using satellite measurements. Low earth orbit (LEO) instruments such as the Advanced Very High-Resolution Radiometer (AVHRR), Moderate Resolution Imaging Spectroradiometer (MODIS), Multiangle Imaging Spectro Radiometer (MISR), Visible Infrared Imaging Radiometer Suite (VIIRS), and Sea-viewing Wide Field-of-view Sensor (SeaWiFS), can provide daily aerosol properties for the global domain (Hsu et al.,2004,2006,2017,2019; Jackson et al., 2013; Jethva et al.,2007; Levy et al., 2013; Lyapustin et al., 2018; Martonchik et al., 2009; Remer et al., 2005). While significant diurnal variations in AOPs have been observed at daily and local scales, emphasizing the importance of geostationary satellite measurements for both air quality and climate studies, the temporal resolutions of LEO satellites (typically 1 day) have limitations in investigating the diurnal variation and transboundary transportation of aerosols (Lennartson et al., 2018; Zhang et al., 2018). Geostationary earth orbit (GEO) instruments such as the Advanced Baseline Imager (ABI), Geostationary Ocean Color Imager (GOCI), GOCI-II, Meteorological Imager (MI), and Advanced Himawari Imager (AHI), have contributed to the operational monitoring of the continuous spatio-temporal variations in AOPs at continental spatial scales with temporal resolutions of minutes to hours using the visible and near-infrared channel (Choi et al., 2018; Kim et at., 2016; Kondragunta et al., 2020; Lee et al., 2023; Yoshida et al., 2018).

Besides spatial and temporal resolutions, another critical consideration for satellite aerosol retrievals is channel specification. Every above-mentioned instrument except GOCI-II uses only visible (Vis) and near-infrared channels. However, near-ultraviolet (UV) spectral region uniquely leverages its sensitivity to aerosol absorption. Thereby, it provides valuable insights into aerosol optical properties. A major advantage of near-UV measurements is that the surface reflectance in the near-UV region is darker than that in the visible region. This enables the derivation of AOPs over a bright surface. In addition, observations in the UV region are sensitive to aerosols' radiative absorption and aerosol layer height (ALH) information because Rayleigh scattering is reduced below the aerosol layer owing to aerosol attenuation (Kayetha et al., 2022; Torres et al., 2005).

The Ozone Monitoring Instrument (OMI) serves as an example of an LEO sensor that utilizes UV wavelengths for aerosol retrievals. It has measured radiances in the 270–500 nm spectral range and offered global coverage at a spatial resolution of $13 \times 24$ km at nadir since 2004 (Levelt et al., 2018). OMI employs two aerosol algorithms. The first one, OMAERO (Curier et al., 2008), developed and maintained by the Royal Netherlands Meteorological Institute (KNMI), is a multiwavelength algorithm that relies on spectral fitting procedures to derive aerosol properties. The other is the OMI near-UV aerosol retrieval algorithm (OMAERUV). It focuses on retrieving key atmospheric aerosol properties including the aerosol optical depth (AOD), single scattering albedo (SSA), and absorbing aerosol index (AI) (Torres et al., 2007).

The OMAERUV algorithm has its heritage in the Total Ozone Mapping Spectrometer (TOMS) aerosol retrieval algorithm. It uses reflectance measurements at 354 and 388 nm to determine the AOD and SSA using the two channel inversion method (Torres et al., 2002; Torres et al., 2007). The global statistics reported by Ahn et al. (2014) indicate a correlation coefficient (R) of 0.81. However, OMAERUV provides a lower R (0.63) over Central and East Asia (Zhang et al., 2015). In addition, the Tropospheric Monitoring Instrument (TROPOMI) aerosol algorithm (TropOMAER) was developed as an adaptation of OMAERUV. A comparison between Aerosol Robotic Network (AERONET) and TropOMAER AOD at 12 locations yielded an R of 0.82 and a root mean square error (RMSE) of 0.19 (Torres et al., 2020).

The Geostationary Environment Monitoring Spectrometer (GEMS) is the first UV-Vis hyperspectral satellite instrument in a GEO. It is onboard the Geostationary Korea Multi-Purpose Satellite-2B (GEO-KOMPSAT-2B or GK-2B). GEMS was launched on February 19, 2020 (Kim et al., 2020). The objective of the GEMS mission is to monitor the hourly air quality in Asia (5°S–45°N, 75–145°E) with a fine spatial resolution ($3.5 \times 7.7$ km² at Seoul, South Korea). GEMS provides hyperspectral





measurements covering 300–500 nm at a spectral resolution of 0.6 nm. Considering the solar zenith angle (SZA), the GEMS
east–west scan profiles are between morning, noon, and afternoon following the sunlit part of the globe to cover the full field
of regard (FOR). The GEMS aerosol retrieval (AERAOD) algorithm is based on OMAERUV algorithm and the optimal
estimation (OE) method by finding the optimized values of AOD, SSA, and ALH from GEMS measurements at six
wavelengths (354, 388, 412, 443, 477, and 490 nm). In order to overcome the challenge posed by the limited degree of freedom
for signal in the GEMS wavelength range, this algorithm employs the two channel inversion method that is used in the
OMAERUV algorithm to retrieve AOD and SSA. Subsequently, these retrievals are used as the first guesses for the OE method
(Kim et al., 2018). The six wavelengths in the UV-Vis region contain information regarding the aerosol absorption in the UV
region and the absorption bands of the oxygen dimer ($O_2$-$O_2$) at 477 nm. This method was tested using the OMI Level 1 data
and was used to derive key aerosol parameters, including AOD, SSA, ALH, UV, and VisAI (Kim et al., 2018; Go et al., 2020a,
2020b). Kim et al. (2018) reported that a comparison between AERONET and GEMS AOD at 26 locations in Asia yielded an
R of 0.71 and a RMSE of 0.46. The percentage of GEMS SSA within the expected error range of the AERONET inversion
data (±0.03) was denoted by 27.54%. Spectral variations of aerosol absorption in the UV-Vis region were investigated by Go
et al. (2020a) and it is applied to GEMS aerosol algorithm. The GEMS AOD demonstrated a strong correlation with the
AERONET AOD (R = 0.847 and RMSE = 0.285) and the percentage of GEMS SSA within the expected error of ±0.03
increased to 41.64% (Go et al.,2020a). To improve the accuracy of GEMS aerosol retrieval, the use of cloud mask information
and total dust confidence index from MODIS IR channels was tested for synergy (Go et al., 2020b).
However, as the testbed for the GEMS algorithm was on the LEO platform, the time-dependent retrieval bias had not been
observed previously. The diurnal variations in satellite-retrieved AOPs may differ from the actual diurnal variations in the
AOPs. This discrepancy can be attributed to the different patterns of bias observed over time among the different geostationary
satellites and retrieval algorithms (Choi et al., 2018; Lennartson et al., 2018; Wei et al., 2019; Zhang et al., 2020). This diurnal
bias in AOP measurements can originate from various factors such as errors in the surface reflectance assumption used in the
retrieval algorithm, calibration issues in the Level 1 data, or the presence of short light paths at noon (Ceamanos et al., 2023).
To address this issue, an empirical AOD bias-correction algorithm was developed. This algorithm utilizes the lowest AOD
values observed within a 30-day period in conjunction with the background AOD to obtain a smoothed bias curve for each
pixel of the ABI AOD data (Zhang et al., 2020). This approach helps mitigate the impact of diurnal bias in satellite AOD
retrievals to improve the accuracy by removing artifacts from the retrieval. By applying bias correction methods, more reliable
diurnal variations in AOD can be explained. Beyond traditional statistical methods, bias correction methods based on machine
learning have started to be proposed. Model-enforced post-processing correction involves the use of a machine learning-based
model to predict errors in conventional aerosol retrievals (Lipponen et al. 2021, 2022a, 2022b). This method was trained to
learn the relationship between the input parameters of the satellite measurements and the associated retrieval errors. This
approach provides a practical and effective method to enhance the accuracy of aerosol retrieval without requiring extensive
modifications to existing retrieval algorithms. It leverages machine learning capabilities to improve the reliability and precision
of hourly aerosol measurements obtained from GEO satellite observations.
In this paper, we report the first aerosol monitoring results including the AOD, SSA, and ALH derived using the GEMS aerosol
retrieval algorithm. The remainder of this paper is organized as follows: Section 2 of the paper describes the GEMS data and
the aerosol retrieval algorithm. It also highlights the algorithm updates after the GEMS in-orbit test (IOT) period. Section 3
discusses post-process correction for near-real-time retrieval. Section 4 discusses the GEMS aerosol monitoring results for
dust, biomass burning, and absorbing aerosol events over Asia. Section 5 presents an evaluation of the retrieved GEMS AOD,
SSA, and ALH retrievals against AERONET and CALIOP data. Section 6 presents the summary and future work.





2 Data and GEMS aerosol algorithm
2.1 Data description
2.1.1 GEMS normalized radiance
GEMS operation process provides Level-1C (L1C) dataset in purpose of improving the efficiency of Level 2 algorithm process
by combining parameters dispersed in different files into one file. In this study, The aerosol retrieval algorithm used radiances
only with the quality flags of 0 (Good) or 2 (interpolated radiances), determined by the "bad_pixel_mask" variable. Rather
than GEMS irradiance, we used the KNMI solar reference spectrum to calculate the GEMS-normalized radiance (Dobber et
al., 2008). The GEMS irradiance is within the range of -5% to -20% compared with the KNMI solar reference spectrum. It still
requires further improvement in L1 processing. To account for the spectral characteristics of the instrument, the KNMI solar
reference spectrum is convolved with the GEMS spectral response function. GEMS-measured irradiances are planned to be
employed when an improved version of the Sun L1C product is released by the National Institute of Environmental Research
(NIER).
Normalized radiances are defined in the following equation:
$$N_\lambda = \frac{I_\lambda}{ESD \; x E_\lambda} \tag{1}$$
where $I$, $E$, ESD, and $\lambda$ are the GEMS radiance, KNMI solar reference spectrum, earth–sun distance correction factor, and
wavelength (354, 388, 412, 443, 477, and 490 nm), respectively. The spectral radiance and irradiance were spectrally binned
and averaged within ±2.2 nm from each wavelength to enhance measurement signals. Additionally, earth–sun distance
correction was used to calculate the normalized radiance.

2.1.2 AERONET
AERONET is a global ground-based remote-sensing network that measures aerosol optical, microphysical and radiative
properties (Giles et al., 2019; Holben et al., 1998; Sinyuk et al., 2020). The measurement systems use Cimel sun photometers
to measure the solar irradiances at eight wavelengths ranging from 340 to 1020 nm and sky radiances at four wavelengths
ranging from 440 to 1020 nm. The AERONET data provide global aerosol information including the spectral AOD and
inversion products such as the SSA, aerosol size distribution, and refractive index. The uncertainties in AODs are wavelength-
dependent. It is approximately 0.01 (Vis) to 0.02 (Near-UV) in direct sun measurements (Dubovik et al., 2002). The
uncertainties of SSA are ±0.03 when AOD exceeds 0.4 at 440 nm (Dubovik et al., 2002). For the evaluation of GEMS AOD
and SSA data from November 2021 to October 2022, we used AERONET V3 Level 1.5 data for AOD and AERONET V3
Level 1.5 hybrid inversion data for SSA from all sites within the entire GEMS domain.

2.1.3 CALIOP
The CALIOP instrument is a two-wavelength polarization-sensitive lidar on the Cloud-Aerosol Lidar and Infrared Pathfinder
Satellite Observations (CALIPSO) satellite. It was launched on April 28, 2006 (Winker et al., 2009). CALIOP monitors the
global vertical profiles of aerosols and clouds by measuring three signals: the backscatter intensity at 1064 nm and the
orthogonally polarized components of the backscattered signal at 532 nm.
Quantitative scattering information from CALIOP instruments was used as reference data for validating the ALH obtained
from passive sensors (Xu et al., 2017; Xu et al., 2019; Nanda et al., 2020; Park et al., 2023). We used CALIPSO Lidar Level
2 Aerosol Profile V3-41 data to validate the GEMS ALH. CALIOP profiles of the extinction coefficient ($\beta_{ext}$) at the 532 nm





channel were utilized to calculate the CALIOP ALH using the following equation:
$$Z_{aer} = \sum_{i-1}^{n} H(i) \left[ \frac{\beta_{ext}(i)}{\sum_{i=1}^{n} \beta_{ext}(i)} \right] \tag{2}$$
where $\beta_{ext}(i)$ is the CALIOP profile of the 532 nm extinction coefficient at height $H(i)$ and $n$ is the number of layers.

2.2 GEMS AERAOD retrieval algorithm
2.2.1 Aerosol optical properties retrieval algorithm for GEMS
The GEMS AERAOD algorithm produces AOD, SSA, and ALH via the OE method. The preliminary GEMS AERAOD was
developed using OMI L1B normalized radiance (Kim et al., 2018; Go et al., 2020a, 2020b). After the launch, the algorithm
was tested using the GEMS observation during the IOT period, and several parts of the algorithm were updated. This section
briefly describes the GEMS AERAOD algorithm; AERAOD L2 data; and updates including the Look-Up Table (LUT), cloud
masking procedure, surface reflectance estimation, and post-processing after the IOT period. The general flow of the GEMS
AERAOD retrieval algorithm is illustrated in Figure 1.
GEMS algorithm adopts a LUT approach to optimize computation efficiency. The LUT is calculated assuming AOPs of three
aerosol types by using a radiative transfer model (RTM), the Vector Linearized Discrete Ordinate Radiative Transfer code
(VLIDORT) (Spurr, 2006). The AOPs of Highly absorbing fine (HAF), Dust, and Non-absorbing (NA) are integrated from
AERONET inversion data and are applied for the RTM simulation. The details of the updated LUT are described in section
2.1.2. The preliminary algorithm used the OMI climatology Lambertian equivalent reflectance (OMLER v003) datasets as
surface reflectance, but for the GEMS AERAOD algorithm, GEMS L2 surface reflectance at 354, 388, 412, 443, 477, and 490
nm are obtained by minimum reflectance method. The details of surface reflectance estimation are described in section 2.1.3.
The GEMS AERAOD provides UV and visible (Vis) AI to indicate the qualitative radiative absorptivity and particle size
information, respectively (Torres et al., 2002). The GEMS UVAI and VisAI were calculated using the following equations:
$$\text{AI} = -100 \left[ \log \left( \frac{N_{\lambda_1}}{N_{\lambda_2}} \right)_{meas} - log \left( \frac{N_{\lambda_1}(LER_{\lambda_1})}{N_{\lambda_2}(LER_{\lambda_2})} \right)_{calc} \right] \tag{3}$$
where $N_{\lambda_1}$ and $N_{\lambda_2}$ are the normalized radiances at the 354/388 (477/490) nm wavelength pair for UVAI and VisAI,
respectively. The subscripts *meas* and *calc* represent the measured and calculated normalized radiances, respectively.
Aerosol type among HAF, dust and NA is selected using the UVAI and VisAI. The NA type was detected by a negative UVAI
value. The dust and HAF types were distinguished by VisAI. When both AIs were positive, the dust type was selected. Sun
glint and cloud masking leave only the pixels appropriate for aerosol retrieval. The glint mask is set for glint angles less than
35°. The details of the cloud-masking procedure are described in Section 2.1.4. The *a priori* states of AOD and SSA at 443 nm
were obtained by two-channel inversion with neighboring wavelengths (354 and 388 nm) over both land and ocean. The
assumption was that the climatology of ALH was based on CALIOP. The a priori states of the AOD and SSA were supplied to
solve the Levenberg–Marquardt equation. The optimal ALH was retrieved by fitting the normalized radiance between the
measured and calculated values for the OE routine. The details of the GEMS aerosol inversion procedure are described by Kim
et al. (2018).
To improve the accuracy of near real-time GEMS AOD retrieval, a model-enforced post-process correction step was
implemented using a random forest (RF) model. By combining GEMS aerosol retrieval with this post-processing correction
model, more reliable and accurate near real-time AOD estimates can be obtained.






2.1.2 LUT calculation
In this study, the AOPs were considered as described by Kim et al. (2018) and Go et al. (2020a). However, the dimensions of
the LUT varied (as shown in Table 1) compared with Kim et al. (2018). The nodes for the 412 nm SSA node for NA were
added. In addition, the nodes for AOD in the LUT were extended to include the values at 5.0 and 10.0 because the previous
maximum node was 3.6. These modifications enable the retrieval of exceptionally severe aerosol events during GEMS
observations. The preliminary GEMS AERAOD retrieval algorithm utilized the normalized radiance at six specific
monochromatic wavelengths (354, 388, 412, 443, 477, and 490 nm). However, satellite measurements averaged over a specific
wavelength range produce more stable values than measurements obtained at individual monochromatic wavelengths. This
increased stability is attributed to the averaging of random errors (i.e., instrument noise). Consequently, a spectral-binning
LUT approach was employed to reduce random errors and improve the stability of the measurements. This allowed for more
reliable and consistent observations. Compared with monochromatic wavelengths, the spectral binning method is
computationally intensive. Therefore, the calculations were performed using the Mie theory without considering the non-
sphericity of the dust. The process of spectral binning LUT in the GEMS aerosol algorithm involves three steps: 1) A reference
spectrum is generated using an RTM, which provides a spectral interval of 0.1 nm. 2) The calculated spectrum is convolved
with the GEMS spectral response function and resampled to the target spectral grids with a resolution of 0.2 nm. (Kang et al.,
2020). 3) The resampled spectrum is averaged at intervals of ±2.2 nm at six central wavelengths (354, 388, 412, 443, 477, and
490 nm) and saved in the LUT. This range is selected to account for the calculation capacity and reduce the impact of random
errors. During the retrieval process, the GEMS L1C normalized radiances after being averaged at intervals of ±2.2 nm at six
central wavelengths are compared with the calculated spectrum in the LUT. By these steps, the spectral binning LUT aims to
generate more stable retrieval results for aerosol properties.

2.1.3 Surface reflectance estimation
In this study, several improvements were introduced. These include an updated GEMS surface reflectance estimation. The
preliminary GEMS AERAOD retrieval algorithm used the OMI surface reflectance climatology data product OMLER v003
(Kleipool et al. 2008), with a spatial resolution of $0.5 \times 0.5°$. The limitation of the previous surface reflectance data was its
coarse spatial resolution compared with that of GEMS pixels. This resulted in discontinuities in the GEMS AOPs owing to
spatial resolution differences. To address this limitation, the updated GEMS surface reflectance has a finer spatial resolution
$(0.1 \times 0.1°)$. This closely aligns with the GEMS pixel resolution. This enhancement enables a more accurate aerosol retrieval
at the pixel level. The compiled hourly surface reflectance indirectly reflects the bidirectional reflectance distribution function
(BRDF) effect. In addition, a new hourly surface reflectance database was generated using the minimum reflectance method
based on the GEMS data. The algorithm adopts the climatological minimum reflectance method for each pixel over a ±15-day
window spanning a period of two years. Several tests were performed to evaluate different time windows and methods for
constructing accurate surface reflectance. These tests evaluated the effectiveness of using a ±15-day window as well as
alternative options such as a previous 30-day window. In addition, different methods including the minimum reflectance and
second minimum reflectance approaches were evaluated to determine the most suitable one for generating appropriate surface
reflectance values (not included in this study).
The background AOD (BAOD) was considered in the retrieval algorithm. The BAOD represents the baseline level of AOD
that is consistently present in a region. Recent studies have shown that incorporating BAOD into an algorithm can reduce the
uncertainty associated with satellite-based AOD remote sensing (Kim et al., 2014, 2021). Zhang et al. (2016) estimated BAOD
as the lowest fifth percentile of AERONET AOD over a two-year period and improved the performance of the VIIRS aerosol





algorithm. It has been observed that Asia experiences relatively high BAOD values with seasonal variation. For example, at
the Dhaka University site, the monthly BAOD over the past two years varied from a minimum of 0.124 in August to a
maximum of 0.685 in April. Therefore, considering the seasonal variation in BAOD for atmospheric correction can help
mitigate the uncertainty in satellite-derived AOD retrieval, particularly over Asia. The monthly BAODs were calculated using
the following equation for each $0.1 \times 0.1°$ box from November 2020 to October 2021:
$$\tau_{grid,b,m}(lat, lon) = \sum_i W_i \tau_{b,m,i}, \sum_i W_i \tag{4}$$
where $\tau_{grid,b,m}(lat, lon)$ is the interpolated BAOD 443 nm at *(lat, lon)* in month *m*. $W_i$ is the inverse distance weighting
function, which is defined as $e^{-d_i(lat,lon)/d_0}$. $d_i(lat, lon)$ is the distance between the AERONET site and GEMS pixel and $d_0$
is a constant, respectively. $\tau_{b,m,i}$ is the lowest fifth percentile of AERONET AOD over a two-year period at AERONET site *i*
in month *m*.
Figure S1 shows the monthly BAOD obtained based on the AERONET AOD data. Additionally, the fifth percentiles of the
AERONET AOD 443 nm values at each AERONET site are plotted as circles for reference. It is evident that regions such as
India exhibit a high BAOD of over approximately 0.15 throughout the year, regardless of the month. However, seasonal
variations in BAOD occur over the Indochinese Peninsula, Korea, and China. These areas experience heavy pollution from
biomass burning during the dry season and dust events from deserts. Both these contribute to increased atmospheric aerosol
concentrations. These enhancements, including the use of hourly GEMS surface reflectance and incorporation of monthly
BAOD, can result in improved aerosol retrieval.

2.1.4 Cloud masking procedure
The GEMS aerosol algorithm retrieved AOPs only in cloud-free pixels. Clouds exhibit spatial inhomogeneity and higher
brightness than aerosols. This study aimed to enhance the cloud-masking process in the GEMS aerosol algorithm by addressing
the limitations of previous simple cloud-masking techniques. The previous method relied on a (1) fixed threshold for
reflectance at 412 nm and (2) standard deviation test of reflectance within a $3 \times 3$ pixel area. To improve the performance of
cloud masking, an additional cloud removal technique has been introduced in this study. These tests include the following: (3)
470/477 nm normalized radiance ratio test. It involves a threshold test for the ratio of the normalized radiance values at 470
nm and 477 nm. This contrasts the presence of clouds using absorption bands of $O_2$-$O_2$. (4) The difference between hourly
surface reflectance database and the calculated scene reflectivity at 412 nm: Significant differences indicate the presence of
clouds (Torres et al., 2013). (5) Standard deviation test of normalized radiance at 477 nm within a $3 \times 3$ pixel area: The
threshold for this test can vary based on the latitude considering the regional differences in cloud characteristics. (5-1) Standard
deviation in $3 \times 3$ pixel > *f(latitude)* (5-2) after 3-1, standard deviation in $3 \times 3$ pixels > *f(latitude, number of cloud pixels*
*detected method (1), (3), (4) in $3 \times 3$ pixels)*. A final cloud mask was applied after the aerosol retrieval. This included (6)
filtering out high AOD values using a threshold that is a function of the number of cloud pixels detected by methods (1), (3),
(4), and (5) in $11 \times 11$ pixel over the ocean (Lyapustin et al., 2021). This helps remove residual clouds. By implementing these
new methods, the algorithm aims to improve the effectiveness of cloud detection and removal in GEMS pixels.

3 GEMS post-process correction for the near-real-time retrieval
The GEMS AOD exhibited a diurnal bias pattern that fluctuated throughout the day. It formed a U-shape, with a minimum at
03:00 UTC (as will be demonstrated in Section 5.1). To improve the accuracy of near real-time GEMS AOD retrieval, a model-
enforced post-process correction step was implemented using a random forest (RF) model proposed by Lipponen et al. (2021).





This concept was trained to learn the relationship between the hourly GEMS data and AOD errors (GEMS-AERONET AOD)
and to predict the AOD errors at the target time. To enable near real-time retrieval, the proposed method consists of two main
parts: modelling and prediction. In the modelling part, the input data for the RF model includes GEMS data (normalized
radiances at six wavelengths, scattering angle, viewing zenith angle (VZA), relative azimuth angle (RAA), SZA UV and VisAI,
aerosol type, AOD, and clear fraction (ClearFrac) (which is the ratio of clear-sky pixels to the total pixels within the 0.25°
radius from the pixel center)). The data also include auxiliary information such as time, land–sea mask, and elevation. The
target data for training were the AOD errors. Each of these was calculated as the difference between the GEMS AOD and
AERONET AOD at the corresponding single GEMS pixel where the AERONET site was located. The predictors and target
variables were collected for a time window ranging from N days to one day before the target time. After conducting several
tests, N was determined to be 30 days. In the prediction part, the input variables including the GEMS data and auxiliary
information in the target time were used for the pretrained RF model. Using these inputs, the model predicted the error in the
GEMS AOD in near real-time. This predicted error value was then applied to the first retrieved GEMS AOD from the retrieval
algorithms. This resulted in the production of the post-processed GEMS AOD.
In addition, the diurnal bias pattern in the GEMS SSA also exhibited fluctuations throughout the day, forming a bell shape with
a minimum at 03:45 UTC. This is shown in Section 5.2. The post-processing method adopted was similar to that used for AOD.
This method was trained to determine the relationship between hourly GEMS data and SSA errors (the difference between
GEMS at 443 nm and AERONET SSA at 440 nm) and predict SSA errors for the target time. The key difference between the
RF model predicting the AOD error and that predicting the SSA error is as follows: the second model includes the GEMS SSA
as an input variable, and then, 19 input parameters are used to construct the RF model.
Unlike AOD and SSA, the postprocessing of ALH using an RF model is inherently limited. CALIOP is predominantly used as
reference data for ALH. Because CALIOP is an LEO satellite, pixels co-located with GEMS ALH data are available only from
03:45 to 07:45 UTC. This renders it inaccessible as a reference hourly dataset covering 22:45–02:45 UTC. Unlike AEROENT,
the use of data from ground-based lidar is severely constrained by the limited number of observation stations and restricted
geographical areas in which lidars are deployed.

4 Aerosol events
4.1 Dust aerosol event (2022.04.08)
Figure 2 present an example of hourly maps of the GEMS aerosol product including AOD, SSA, ALH, UVAI, and VisAI for
April 8, 2022. These results are the GEMS AOD and SSA before post-processing. The selected case is for the dust aerosol
event over northwestern China. The GEMS false RGB is shown using R (477 nm), G (412 nm), and B (354 nm) bands similar
to those of the OMI false RGB method (Levelt et al., 2006).
As shown in Figure 2, different retrieval regions with respect to time are shown as the GEMS scan profile varies with the SZA.
Overall, the GEMS AOD shows a significantly good agreement with the AERONET AOD measurements. It captures higher
values in the Beijing–Hebei–Tianjin (BTH) region and lower values over South Korea and Japan. High GEMS AOD values
were evident along the dust plume, attaining two at 06:45 UTC. In the case of SSA, the retrieval results demonstrated a
relatively lower accuracy (notably in the BTH region) compared with AOD. In general, from 22:45 to 05:45 UTC, the SSA
values displayed good concordance with both AERONET and GEMS SSA. However, from 06:45 to 07:45 UTC, the SSA
numbers did not match over Beijing. Compared with the Beijing region, the results are more consistent in the dust plume. The
SSA values remained relatively stable at approximately 0.92–0.96 over time. However, the GEMS SSA tended to have a
positive bias compared with the AERONET values. This is shown in Section 5.2. The GEMS ALHs were ~3–4 km for the dust



plume over the Taklamakan Desert and ~1.0 km over the Beijing region. The GEMS ALH exhibited continuous spatial and temporal patterns. The UVAI provides information regarding the radiative absorption of aerosols. It attained a maximum of four for dust plumes, thereby indicating significant aerosol absorption. However, over Beijing, the SSA was ~1. This indicated a marginal absorption owing to the different aerosol emission source. VisAI provides information on the aerosol size. In regions with a dust plume, the VisAI value was higher than that in the background areas. This indicated the presence of coarse aerosol particles.

4.2 Biomass burning event (2022.03.19)

Figure 3 illustrates maps of the GEMS aerosol product at 06:45 UTC on March 19, 2022. It represents a biomass-burning event over mainland Southeast Asia. These results were obtained for the GEMS AOD and SSA before post-processing. During the dry season in this region, highly absorbing fine pollution particles are prevalent (Yin et al., 2019). The GEMS AOD > 1.6. This indicated a significant aerosol loading and enhancement during the event. The GEMS SSA was ~0.88. This indicated aerosol absorption during this event. The ALH ranged from 2 to 3 km within the biomass-burning plume. The GEMS ALH was not retrieved along the east-to-west straight line at ~22.5 °N, which are bad pixels in the CCD. The GEMS UVAI showed hotspots and fine features associated with this event. Thus, it captured aerosol absorption in the ultraviolet spectrum. VisAI exhibited higher values than the background. This case study demonstrates that the GEMS provides valuable insights into aerosol properties during specific events such as biomass burning, and can capture temporal and spatial variations in AOD, SSA, ALH, UVAI, and VisAI.

Figure 3g shows a comparison of the CALIOP extinction coefficients at 532 nm, the CALIOP ALH, and the GEMS ALH over the CALIOP path (the green line on the GEMS false RGB image in Figure 3a). Figure 3g illustrates a clear relationship between the GEMS AOD and accuracy of GEMS ALH. The accurate retrieval of ALH requires the presence of a sufficient amount of aerosols in the atmosphere. GEMS ALH closely follows the latitudinal variation in CALIOP ALH. As the latitude increased from 18° to 21°, the GEMS ALH followed the CALIOP ALH and showed an increase in altitude. In the latitude range of 24°–28°, the GEMS AOD decreased, and the GEMS ALH exhibited scattered variations owing to weaker signals. In the scatter plot comparing CALIOP ALH and GEMS ALH (Figure 3h), 39.88% of the pixels are within the expected error range of 0.5 km, and 68.10% of the pixels are within the expected error range of 1 km. As the GEMS AOD values decreased, the GEMS ALH pixels were more likely to be outside the expected error range.

4.3 Absorbing aerosol event (2021.12.04, 2021.12.23)

Figure 4 shows an example of the GEMS AOD before and after post-processing for an absorbing aerosol case over Indo-Gangatic Plane (IGP) at 04:45 UTC on December 4, 2021. During the wintertime in this region, atmospheric haze is prevalent (Ram et al., 2012). Recent studies have shown that primary aerosols and precursors for secondary aerosols emitted from fossil fuel combustion and biomass burning are released into the atmosphere (Singh et al., 2021). Figure 4a shows the GEMS false RGB image with AERONET stations represented by circles. The color indicates the AERONET AOD. Two distinct aerosol plumes are observed. The northwest shows an AOD of ~0.8, whereas the southeast has a value of ~1.3. Figure 4b shows the GEMS AOD data. The spatial distribution of the GEMS AOD is similar to that of the AERONET AOD in Figure 4a. However, the values are marginally lower than those of the AERONET AOD. Meanwhile, the AOD increased after post-processing, particularly in the moderate AOD range (~0.7). Moreover, the GEMS AOD was closer to the AERONET AOD (Figure 4c). Specifically, at the Gandhi_College site (25.871 °N, 84.128 °E) and Lahore (31.480 °N, 74.264 °E), postprocessing resulted in more reasonable values.



Figure 5 shows the maps of the GEMS SSA and the GEMS SSA after post-processing for an absorbing aerosol case over India,
Bangladesh, and mainland Southeast Asia at 03:45 UTC on December 23, 2021. Figure 5a shows the GEMS false RGB image
with AERONET stations represented by circles. The color indicates the AERONET SSA at 440 nm. The AERONET SSA
values are ~0.9 in India and Bangladesh, and ~0.93 in Thailand. Before postprocessing, the GEMS SSAs exhibit values of
~0.96 in the Indian region and ~1.0 in the other areas. However, following postprocessing, the GEMS SSA values converged
to be more similar to the AERONET SSA values. Nonetheless, a marginal tendency for overestimation remained.

5 Validation in GEMS AERAOD product
This section evaluates the GEMS AOD and SSA at 443 nm according to the aerosol type and measurement time using the
AERONET data in the entire GEMS domain. We used AERONET version 3 level 1.5 data to validate both AOD and SSA to
ensure a larger dataset for validation purposes. Figure 6 illustrates a map of the AERONET sites used for GEMS AOD and
SSA validation, in conjunction with site-specific data counts. The AERONET AOD data generally showed higher counts for
South Korea, China, and Taiwan. Meanwhile, sites in South and Southeast Asia typically had fewer data points. Similarly, the
number of AERONET SSA data points showed a distribution similar to that of AOD. However, AERONET sites #38, #39, and
#47 in India had over 400 validation points. In addition, we retrieved the GEMS ALH and compared it with the CALIPSO
level 2 extinction coefficient profiles at 532 nm as well as with the CALIOP ALH defined by Equation (2).

5.1 Aerosol optical depth
In this section, the GEMS AOD at 443 nm is validated against AERONET data across the entire GEMS domain from November
1, 2021 to October 31, 2022. The GEMS AOD data were spatially collocated within a 0.25° radius of the AERONET stations
and temporally within a 30 min window of the GEMS measurement time. When a specific aerosol type in the GEMS was
present in more than 90% of the pixels within the validation radius, an aerosol type validation was conducted.
Figure 7 presents the results for all the pixels and each aerosol type (HAF, dust, and NA). The statistics include R, RMSE,
mean bias error (MBE), slope, y-offset, Q value indicating the percentage of data points within the maximum (0.1 or 30%
AOD) error range, and the Global Climate Observing System (GCOS) requirement (defined as the maximum (0.03 or 10%
AOD)). The total GEMS AOD demonstrated a good correlation with the AERONET AOD, with an R of 0.792, RMSE of 0.227,
and MBE of 0.038 (Figure 7a). The Q value was calculated to be 54.84%, with 18.39% of the AOD satisfying the GCOS
requirements. However, the slope and y-intercept were 0.589 and 0.193, respectively. This indicated an overestimation for a
low AERONET AOD and an underestimation for a high AERONET AOD. In the case of a low AERONET AOD, there is
evidence of cloud contamination effects. These result in an overestimation of the retrieved GEMS AOD.
The validation shows the differences by aerosol type. The HAF type showed the highest R and Q values compared with the
other aerosol types (Figure 7b). Pixels that deviated beyond the error range owing to the GEMS AOD underestimation were
notably observed in two main categories: sites in the Indian region (which still showed bias notwithstanding the consideration
of BAOD) and sites located in Beijing with an AERONET AOD of approximately 2.0 and a GEMS AOD of approximately
1.0. Among the three aerosol types, the dust type had the fewest samples, accounting for 1 / 10 of the NA (Figure 7c). The R-
value was 0.786, and the slope was the highest among the three types. Pixels that deviated beyond the error range owing to
GEMS AOD underestimation were primarily observed in the Indian region. In contrast, pixels exceeding the error range owing
to GEMS AOD overestimation were located in Northeast Asia. Currently, GEMS uses the same aerosol model (number-size
distribution parameters and real refractive index) over the entire domain for each aerosol type. However, given the varying
bias patterns observed in the dust type, it is necessary to consider regional variations in the GEMS aerosol model (and thus,
the LUT) in future studies. The NA type was selected most frequently among the three aerosol types (Figure 7d). Figure 7d





shows that a significant number of pixels are influenced by cloud contamination, which is particularly evident in regions with
low NA AOD values. It appears that the GEMS aerosol cloud masking process requires further improvement, particularly over
the ocean. The current cloud-masking process may not effectively distinguish small clouds (i.e., broken clouds) near equatorial
regions. This results in an overestimation of the AOD owing to cloud contamination. This phenomenon has been observed
frequently at AERONET stations located near the equator. The underestimation of high AOD values in the GEMS aerosol
algorithm can be attributed to the effect of the current aerosol model assumption used in the algorithm. This emphasizes the
importance of understanding the AOPs to better characterize these in the atmosphere, particularly in the UV region.

Figure S2 and Table 2 present the hourly AOD validation results and statistical metrics including N, R, slope, y-intercept,
RMSE, MBE, Q value, and GCOS. It is important to note that the GEMS varies its E-W scan profile depending on the SZA.
Therefore, the sites used for validation may not have remained consistent over time. For example, the AERONET stations
around 22:45 UTC and 23:45 UTC were mostly used for validation in the eastern region of GEMS, whereas those around
06:45 UTC and 07:45 UTC were expected to be in the western region of GEMS. A systematic error analysis is planned in a
future study. Nevertheless, the hourly validation results of the GEMS AOD provide significant insights. The hourly slopes of
the GEMS AOD exhibited a diurnal variation, starting at 0.730 at 22:45 UTC; decreasing to 0.534 and 0.555 by 1:45 UTC
and 2:45 UTC, respectively; and subsequently increasing to 0.647 and 0.617 at 06:45 and 7:45 UTC, respectively. However,
the R-values remained relatively stable over time. Most time intervals exhibited R values of approximately 0.77 or higher
except for 22:45. Figure S2 and Table 2 show that the diurnal variation in GEMS AOD did not precisely reflect the actual
diurnal AOD variation. Thus, it is necessary to correct and produce a consistent dataset over time to investigate the diurnal
variations in aerosol properties. A machine learning model using RF was used to train the hourly dependent error
characteristics, remove artifacts in the retrieval processes, and maintain the physical signals.
Figure 8a shows the comparison results for GEMS AOD after model-enforced post-processing correction with AERONET
data. For near-real-time post-processing correction, data from the past 30 days were used for training. Therefore, these results
were evaluated over 11 months: from December 1, 2021, to October 31, 2022. Figure 8a shows that all the statistical metrics
improved. In particular, the slope was closer to one at 0.809, and the y-intercept was closer to zero at 0.068. Additionally, R,
RMSE, and MBE were 0.899, 0.159, and -0.005, respectively. The Q value and GCOS requirements also improved to 79.13%
and 36.08%, respectively. The bias near low AOD values of approximately zero was reduced significantly. Furthermore, the
high AOD values were closer to the 1:1 line. Figure 8b shows the bias of the GEMS AODs before and after post-process
correction with respect to time for all the AOD pixels. After applying the model-enforced post-process correction to the GEMS
AOD data, significant improvements in bias were observed over the diurnal cycle. The original GEMS AOD exhibited an
hourly-dependent bias characteristic. It formed a U-shape with a minimum value near noon, 03:45 UTC. However, with the
implementation of the model-enforced post-processing correction, the diurnal bias was mitigated effectively. This resulted in
a bias value close to zero throughout the day and a decreased standard deviation. Figure 8c illustrates the diurnal variation in
the bias of a low AOD (AERONET AOD < 0.4). The GEMS AOD (red circles) exhibited a positive bias of ~0.1. It was mostly
corrected to values close to zero after post-processing (blue circles). However, certain positive bias was observed at
approximately 22:45 and 23:45 UTC, and at 06:45 and 07:45 UTC. Figure 8d shows the diurnal variation in the bias of high
AOD (AERONET AOD > 0.4). The diurnal variation in GEMS AOD (red circles) shows a clear U-shaped pattern with a
maximum negative bias of approximately -0.2 at 0.3 UTC. However, after post-processing, the bias was still negative but less
than -0.1, which is significantly closer to zero. By incorporating the predicted error, we obtained an improved GEMS AOD
that considers the uncertainties and biases inherent in the retrieval process. This approach helps reduce these biases, including
a low AOD overestimation, high AOD underestimation, and artificial diurnal bias in near-real-time AOD retrievals. The
reduction in artifactual diurnal bias is crucial for ensuring the reliability of hourly GEMS AOD data. This is because it





eliminates time-dependent discrepancies and provides a more representative hourly aerosol distribution. Users can now rely
on corrected GEMS AOD data for various applications without being influenced by diurnal variations in the original
measurements. Variable importance analysis for the post-processing correction of the GEMS AOD was conducted (Figure S3).
GEMS AOD was the most important variable, emphasizing its direct influence on the correction process. VZA and elevation
exhibited high importance. However, their significance can be attributed not only to their inherent properties but also to their
role in conveying AERONET location-related information. Aerosol type appeared to have less significance in the RF models.
This result contrasted with the notable importance of GEMS UVAI and VisAI. This discrepancy can originate from inaccurate
aerosol type classification in the GEMS aerosol algorithm.

5.2 Single-scattering albedo
This section presents a comparison of the GEMS SSA at 443 nm with the AERONET SSA at 440 nm in the entire GEMS
domain. The validation period and collocation criteria for the AERONET sites were identical to those for the GEMS AOD.
Similar to the AOD, when a particular aerosol type in the GEMS was detected for over 90% of the pixels within a 0.25° radius,
we performed aerosol-type validation. Figure 9 and Table 3 display the validation results for all pixels and each aerosol type.
The statistics including N and percentages are within the expected error ranges (0.03 and 0.05). The uncertainty of SSA is
±0.03 when AERONET AOD 440 nm is over 0.4 (Dubovik et al., 2002). The gray dashed lines indicate an uncertainty envelope
of ±0.03 in SSA, whereas the black dashed lines indicate an uncertainty envelope of ±0.05 in SSA. These reference lines help
assess the agreement between the GEMS SSA and AERONET data within a reasonable error range. When aerosols are not
abundant in the atmosphere, capturing SSA signals from satellite observations is challenging. Therefore, for validation,
separate analyses were conducted for the cases where the GEMS AOD > 0.4 (indicated by the red open circles) and the GEMS
AOD was > 1.0 (indicated by the blue open circles). Notwithstanding the large uncertainties associated with the satellite
measurements, the GEMS aerosol product showed a good overall agreement with the AERONET SSA. When GEMS AOD
exceeds 0.4, the percentage of GEMS SSA within the expected error range of ±0.03 is denoted by 42.76%, and that within the
expected error range of ±0.05 is denoted by 67.25%. When the aerosol signal is strong (when GEMS AOD exceeds 1.0), the
percentage of GEMS SSA within the expected error of ±0.03 (0.05) increases to 56.61% (83.70%). However, the percentage
within the expected error range and scatter plots varied depending on the aerosol type. For the HAF type, the SSAs showed
the largest spread. This indicated a lower accuracy. It was likely to be a result of an ineffective aerosol-type selection (red
circles). However, when AOD exceeds 1.0 (blue circles), these tend to approach the 1:1 line. Moreover, the percentage falling
within the expected error range of ±0.03 increases significantly. For the dust type, the GEMS SSA exhibited a positive bias of
approximately 0.04 compared with the AERONET SSA (red circles). Similarly, when the AOD exceeds 1.0, these biases
decrease, approaching the 1:1 line (blue circles). However, the systematic bias observed in the GEMS SSA for the dust type
indicates the need to refine the assumed dust AOPs in the LUT. The NA type in GEMS was observed to have a significantly
low variability compared with AERONET SSA. The GEMS SSAs showed values close to one compared with the AERONET
data. According to Lee et al. (2010), the NA type is identified when the SSA is above 0.95. However, many NA-type pixels
were observed, with AERONET SSA values below 0.95 in the NA type. This indicates potential inaccuracies in the
classification of the absorbing and NA GEMS aerosol types. Nevertheless, when the AOD is high (blue circles), these
classification errors tend to decrease. This results in values closer to the AERONET SSA.
Figure S4 and Table 4 present the hourly SSA validation results and statistic metrics including the N and percentage within
the expected error range of ±0.03 (±0.05). The GEMS and AERONET SSA exhibited varying distributions over time. The
difference between the GEMS and AERONET SSA was most significant at 03:45 and 04:45 UTC, with a positive bias. This
difference decreased at 22:45 and 23:45 UTC or 05:45 and 06:45 UTC (Figure S4). Similar to the GEMS AOD, the GEMS
SSA showed diurnal variations. These are also reflected in the EE% values shown in Table 4. At 22:45 and 23:45 UTC, the



percentage within the expected error range of ±0.03 exceeded 60%. However, it reduced to less than 30% at 03:45 and 04:45
UTC before increasing again. Further studies are required to understand the bias and accuracy variations in the SSA and
improve the retrieval results. This can also be attributed to the shorter path length in the observation geometry when the
influence of surface reflectance increases, similar to that in AODs.
Figure 10a presents the comparison results for the GEMS SSA after post-process correction and the AERONET data. The
near-real-time post-process correction utilized data from the preceding 30 days for training. The validation period was from
December 1, 2021, to October 31, 2022. Notably, all the statistical metrics demonstrated improvements. Specifically, the
percentage of GEMS SSA falling within the expected error range of ±0.03 was recorded at 68.33%, whereas the percentage
within the range of ±0.05 was indicated at 88.86%. Furthermore, the SSA values exhibited a closer alignment with the 1:1
line. Figure 10b depicts the difference between the GEMS and AERONET SSA over the measurement time. Notably, the
bias pattern observed in the GEMS SSA exhibits artifactual characteristics, thereby forming a bell-shaped curve. In
particular, during the time interval from 01:45 to 05:45 UTC, the mean bias of GEMS SSA consistently surpassed the
expected error range of ±0.03. However, the implementation of model-enforced post-process correction was demonstrated to
be highly effective in mitigating this artificial diurnal bias. This correction methodology resulted in a significant
improvement in the GEMS SSA values within the expected error range. Thereby, it enhanced the overall accuracy of the SSA
retrieval. Variable importance analysis for the post-processing correction of the GEMS SSA was conducted (Figure S5). The
GEMS SSA was the most important variable in the correction process. The GEMS AOD also emerged as a highly influential
variable in the RF models for GEMS SSA post-process correction. Also, aerosol types appeared to have relatively lower
significance within the RF models for SSA correction.
5.3 Aerosol layer height
From November 1, 2021 to October 31, 2022, the GEMS and CALIOP data were co-located for comparison. In this section,
the level-2 aerosol extinction coefficients at 532 nm are used to calculate the CALIOP ALH. This is shown in Equation 2.
GEMS ALH pixels within a 0.05° radius surrounding each CALIOP pixel were averaged and compared with the CALIOP
ALHs within a time window of 1 h from the GEMS observation time. The validation was conducted when the GEMS AOD
values were larger than 0.2. This was because the error in ALH retrieval increased when the presence of aerosols in the
atmosphere was insufficient. Figure 11a shows a histogram of the differences between the GEMS and CALIOP ALH. The total
co-located number of data is 77,318, and the mean difference is -0.225 km. The median of differences is -0.167 km. This
indicates that the histogram distribution of the differences follows a Gaussian distribution although it is skewed marginally in
a positive direction. Figure 11b shows a comparison between the GEMS and CALIOP ALH. These were distributed
predominantly at altitudes less than 2 km. The percentage of data falling within the expected error of ±1 km was 55.3%, and
the percentage falling within the expected error of ±1.5 km was 71.7%. The variability of the GEMS ALH was comparable to
that of the CALIOP ALH.
6 Summary and future work
In this study, we present the first atmospheric aerosol monitoring results from GEMS over Asia. Given that the GEMS
AERAOD algorithm was developed using OMI as input data before GEMS launch, modifications were made considering the
GEO observation characteristics during the IOT period. A new hourly surface reflectance database was created using the
minimum reflectance method with a fine spatial resolution that aligned with the GEMS pixel resolution. In addition, monthly
BAOD maps were incorporated to estimate the hourly GEMS surface reflectance. A new cloud removal techniques
significantly improved effectiveness of cloud detection and enhanced the quality of aerosol retrievals. To avoid discrepancies
between observed and simulated radiance that may arise due to the monochromatic assumption of LUT calculation, we applied
a spectral binning approach to LUT calculation. Finally, post-process correction methods based on machine learning were used
to remove the non-physical diurnal biases in AOD and SSA retrieval. This reduced the biases over time and provided more
reliable hourly GEMS aerosol products in near real-time.
The GEMS aerosol product was investigated for three specific events: dust events over Northeast Asia, biomass burning in
Southeast Asia, and the absorption of aerosols over India. These events highlight the capability of the GEMS to monitor and
provide insights into aerosol properties during various atmospheric events while also emphasizing the importance of post-
processing for data accuracy and agreement with ground measurements.
The GEMS aerosol products were validated against the AERONET and CALIOP data for the entire GEMS domain for one
year (from November 2021 to October 2022). The performance of the GEMS aerosol algorithm was validated to verify its
applicability for studying the distribution of AOPs across Asia. The validation results for each product are summarized below:
GEMS AOD shows a good correlation with the AERONET AOD (R = 0.792). However, it exhibits certain bias patterns.
Notably, an underestimation of AOD in high AERONET AOD and overestimation of AOD in low AERONET AOD occurred
owing to cloud contamination. Different aerosol types exhibited varying validation results: the HAF type with the highest R
and Q values; the dust type with underestimation in India but overestimation in Northeast Asia; and NA type with cloud
contamination issues, particularly for low AOD. This indicated the need for an improvement in the cloud masking process,
particularly over the ocean. Certain deviations beyond the error range of the GEMS AOD were observed in India and Beijing.
The underestimation of the high AOD values can be attributed to the aerosol model. Diurnal variation in retrieval performance
was evident with varying slopes and other comparison statistics throughout the day. As the testbed for the GEMS algorithm
was on the LEO platform, the time-dependent retrieval bias had not been observed previously. Therefore, we adopted a model-
enforced post-process correction and find that this enhances GEMS AOD performance, reducing overall biases. This corrected
data ensures reliability for various applications.
The GEMS SSA at 443 nm was validated against the AERONET SSA at 440 nm over the entire GEMS region. The GEMS
SSA's agreement with the AERONET data was evaluated within a reasonable error range of ±0.03 (±0.05). For GEMS AOD
exceeding 0.4, 42.76 (67.25)% of GEMS SSA is within ±0.03 (0.05) error. This increases to 56.61 (85.70)% for strong aerosol
signals (GEMS AOD above 1.0). However, the accuracy varies among the aerosol types. The HAF type has a higher variability
and lower accuracy. The dust type has a marginal positive bias, particularly when the AOD is high. Similar to AOD, post-
process correction for the GEMS SSA data yielded significant enhancements in statistical metrics.
The GEMS and CALIOP data were then compared. The GEMS ALH was compared with the CALIOP ALH when the GEMS
AOD exceeded 0.2. The results showed a mean difference of -0.225 km, with 55.29% of data being within ±1 km and 71.70%
being within ±1.5 km. The GEMS ALH exhibited variability similar to that of CALIOP ALH.
Several methods can be used to further improve the results of the GEMS aerosol algorithm. First, additional satellite data could
be integrated for cloud detection. Incorporating data from other satellite sensors with IR channels such as the AMI can provide
complementary information for cloud masking. Second, it is necessary to consider the AOPs used in the LUT to improve the
GEMS aerosol algorithm. It is particularly important to incorporate more ground-based observations in the UV region, such
as those from the Pandora Instrument and SKYNET. Collecting ground-based observations in the UV region and incorporating
these into LUT can enhance the performance of this algorithm. Finally, regional LUTs with data from diverse regions that
consider the variability in AOPs based on regional characteristics are crucial. Overall, the improvements to the GEMS aerosol
algorithm contribute to advancing our understanding of aerosol properties and their effects on the environment. Thereby, it



provides valuable information for diverse applications including air quality monitoring, air quality data assimilation, and health
impact assessments in Asia.

*Code availability.* The GEMS L2 AERAOD algorithm is not available publicly.

*Data availability.* GEMS L2 AERAOD was downloaded from the Environmental Satellite Center website
(https://nesc.nier.go.kr/en/html/datasvc/index.do).

*Author Contribution.* YC, JK, SG, and MK designed the experiments. WL and DL provided support for the data collection.
SL, MK, HC, OT, and SP contributed to the algorithm development. YC wrote the manuscript with contributions from all the
co-authors. JK reviewed and edited the manuscript. JK provided support and supervision. All the authors analyzed the
measurement data and prepared the manuscript.

*Competing Interests.* At least one of the (co-)authors is a member of the editorial board of Atmospheric Measurement
Techniques.

*Acknowledgements.* We thank all the principal investigators and their staff for establishing and maintaining the AERONET
sites used in this investigation. The CALIOP V3.41 data were obtained from the NASA Langley Research Center Atmospheric
Science Data Center at https://asdc.larc.nasa.gov/project/CALIPSO. The authors acknowledge the National Institute of
Environmental Research, Korea Aerospace Research Institute, for providing the satellite data, and Professor Myoung-Hwan
Ahn and his research group at Ewha Womans University for providing information regarding the GEMS specifications and
Level 1 data.

*Financial Support.* This work was supported by a grant from the National Institute of Environment Research (NIER), funded
by the Ministry of Environment (MOE) of the Republic of Korea (NIER-2023-04-02-050).




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

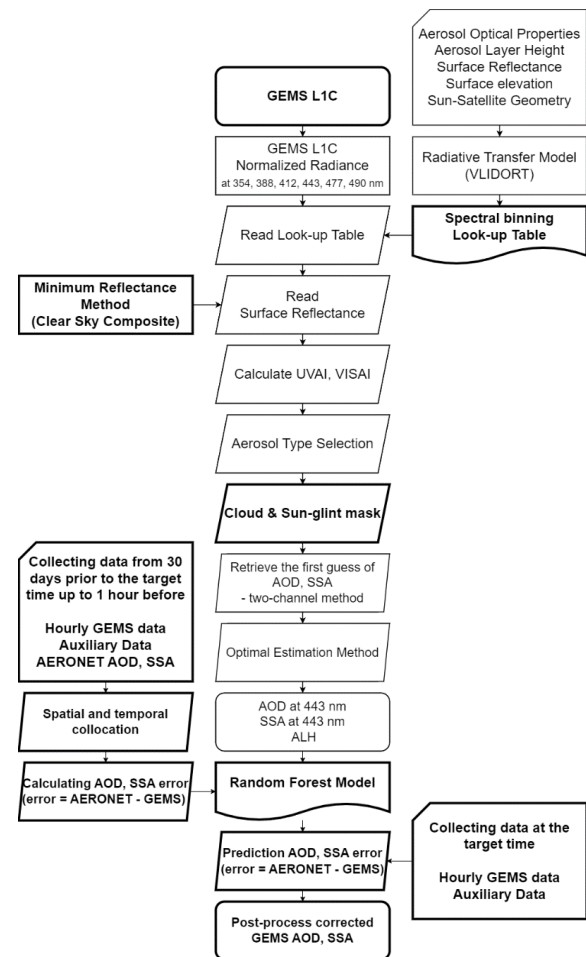


**Figure 1: The flowchart of the GEMS AERAOD retrieval algorithm and the modifications in the study (in bold boxes)**





**Figure 2: Hourly GEMS aerosol products for the dust case on April 8, 2022 over northwestern China. Time-series maps of AOD, SSA, ALH, UVAI, and VISAI from 22:45 to 07:45. The circle denotes an AERONET station, and the filled color indicates the AERONET AOD and SSA at 443 nm in the AOD and SSA columns. GEMS SSA, and ALH are displayed only when GEMS AOD > 0.2.**



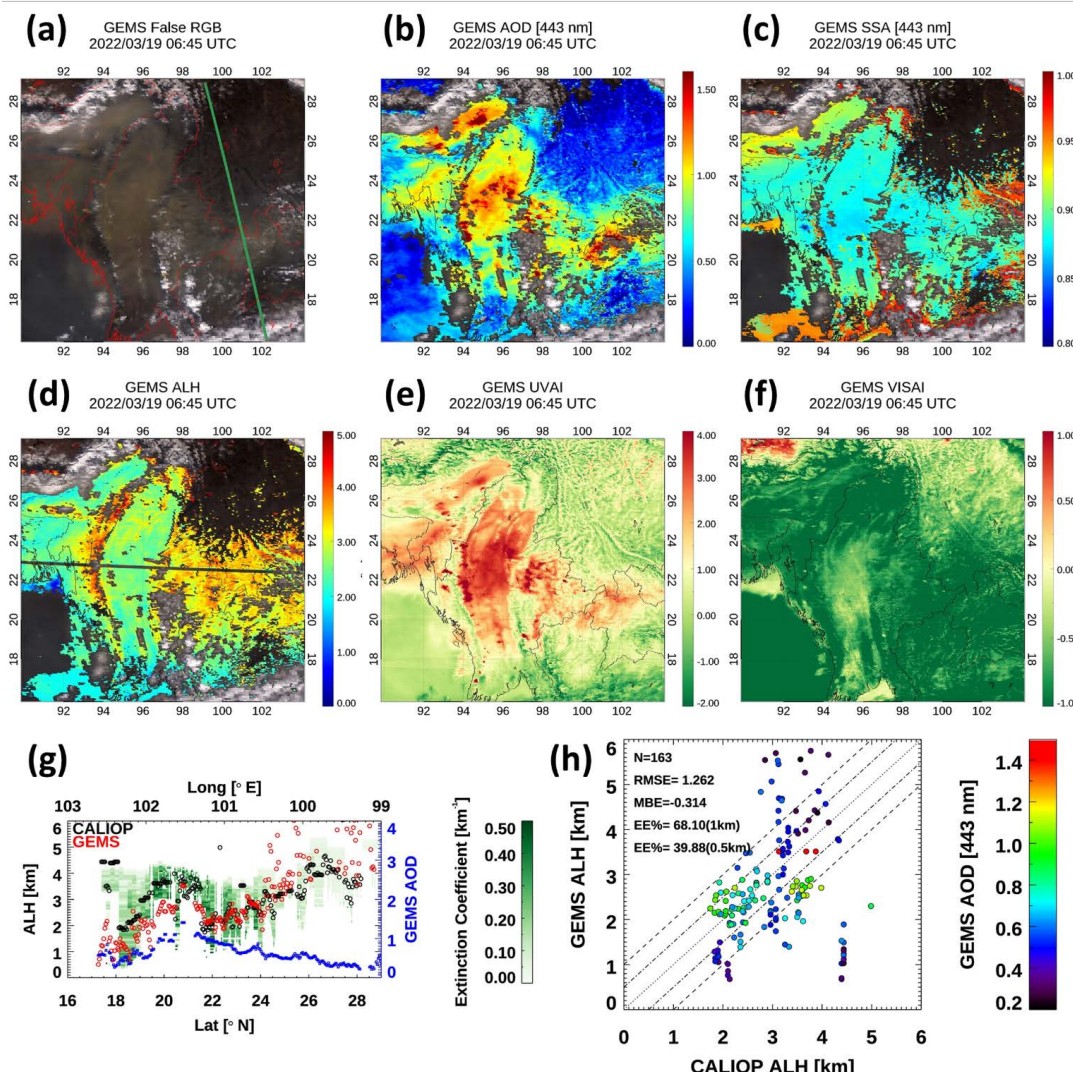

**Figure 3: The example of GEMS aerosol products for biomass burning over mainland Southeast Asia. The maps of (a) GEMS False RGB, (b) AOD, (c) SSA, (d) ALH, (e) UVAI, and (f) VisAI. The green line in GEMS False RGB indicates the overpass path of CALIOP. The GEMS SSA and ALH are displayed only when the GEMS AOD is over 0.2. (g) GEMS ALH compared with CALIOP extinction coefficient in the domain. The background color represents the CALIOP extinction coefficient. The black open circles denote the CALIOP ALH, whereas the red open circles represent the GEMS ALH. The blue squares represent the GEMS AOD. (h) Comparison of GEMS and CALIOP ALH when GEMS AOD > 0.2. The dashed and dash-dotted lines indicate an uncertainty envelope of ±1 km and ±0.5 km in ALH, respectively. The dotted lines represent the 1:1 line. The color in the circles represents the GEMS AOD.**





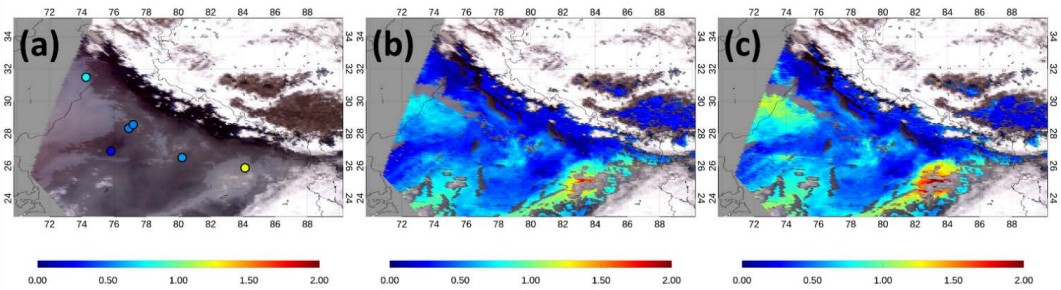


**Figure 4: The example of the GEMS AOD before and after post-processing for an absorbing aerosol case over Indo-Gangatic Plane**
**at 04:45 UTC on December 4, 2021. (a) GEMS false RGB. The circle denotes an AERONET station, and the filled color indicates**
**the AERONET AOD at 443 nm, (b) GEMS AOD and (c) GEMS AOD after post-process correction.**

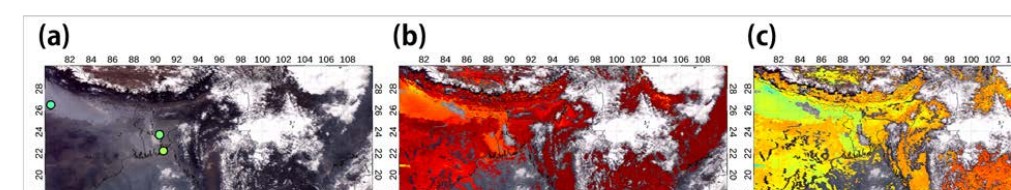


**Figure 5: The example of GEMS SSA and the GEMS SSA after post-processing for an absorbing aerosol case over India,**
**Bangladesh, and mainland Southeast Asia at 03:45 UTC on December 23, 2021. (a) GEMS false RGB. The circle denotes an**
**AERONET station, and the filled color indicates the AERONET SSA at 440 nm, (b) GEMS SSA, and (c) GEMS SSA after post-**
**process correction.**



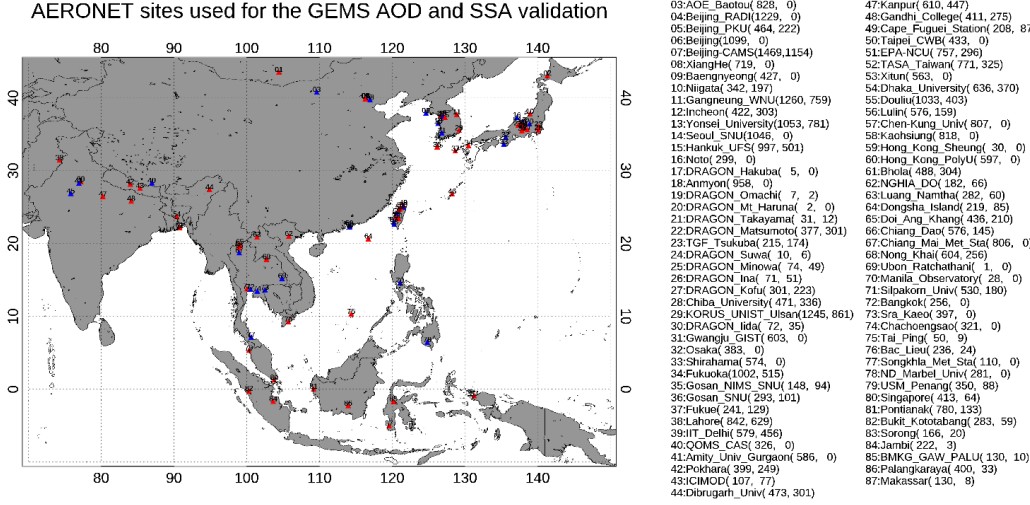

**Figure 6: AERONET sites used for the GEMS AOD and SSA validation. The red color indicates the site where validation points exist for both AOD and SSA. The blue color indicates the site where validation points exist only for AOD. The list of station names in conjunction with the number of AERONET AOD and SSA data points for validation at each station.**

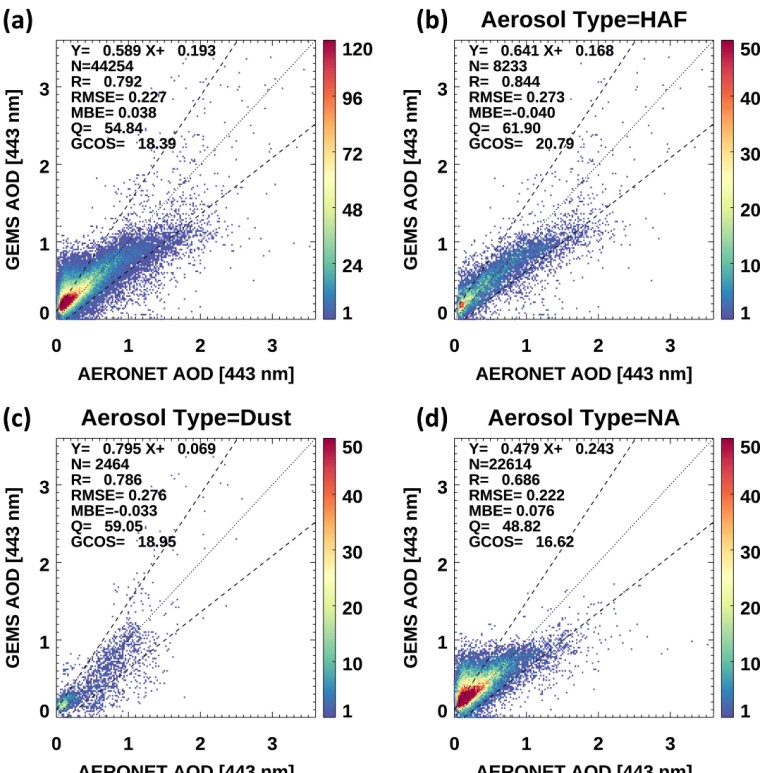

**Figure 7: Comparison of GEMS and AERONET AOD for (a) total and individual aerosol types: (b) HAF, (c) dust, and (d) NA. The dashed lines indicate an uncertainty envelope of maximum (0.1 or 30%) in AOD. The dotted lines represent the 1:1 line. Data from November 1, 2021 to October 31, 2022 are used for comparison.**






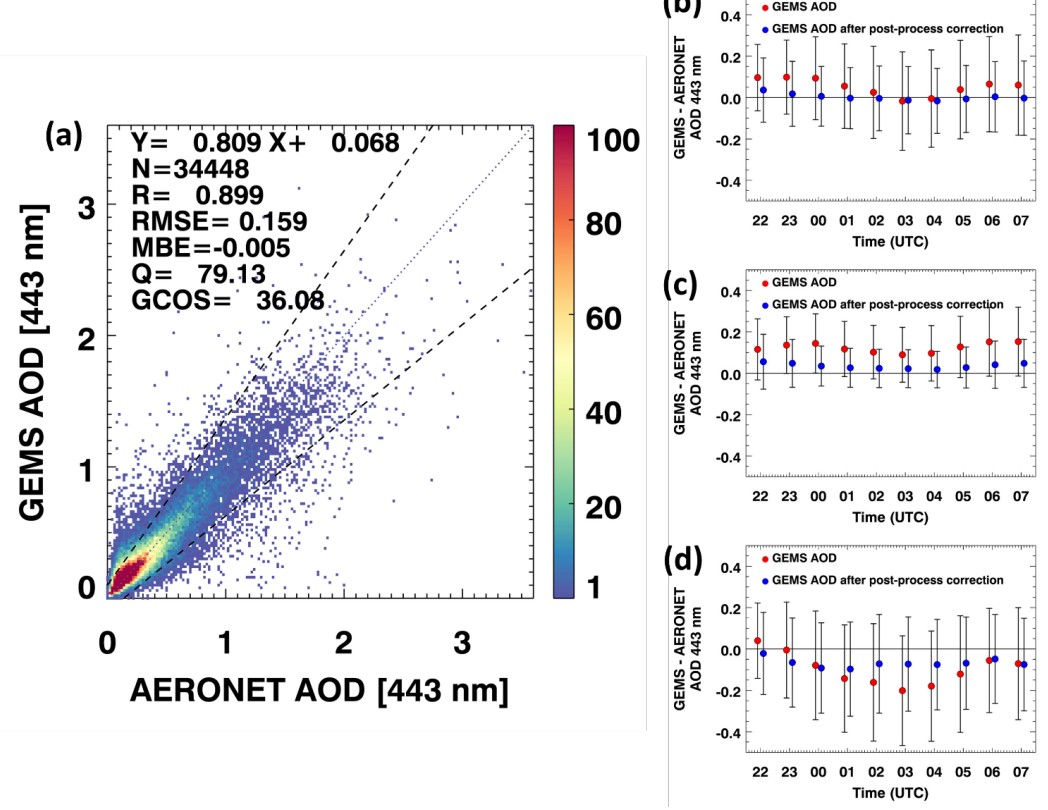


**Figure 8: (a) Comparison of GEMS AOD after post-process correction by machine learning and AERONET AOD. The dashed**
**lines indicate an uncertainty envelope of a larger 0.1 or ±30% in AOD. The dotted lines represent the 1:1 line. The difference**
**between GEMS AOD and AERONET AOD in terms of time. (b) All pixels, (c) pixels when AERONET AOD < 0.4, and (d) pixels**
**when AERONET AOD > 0.4. The red circles represent the GEMS AOD, and the blue circles represent the GEMS AOD after post-**
**process correction. The error bars correspond to the standard deviation. Data from December 1, 2021 to October 31, 2022 are used**
**for comparison.**



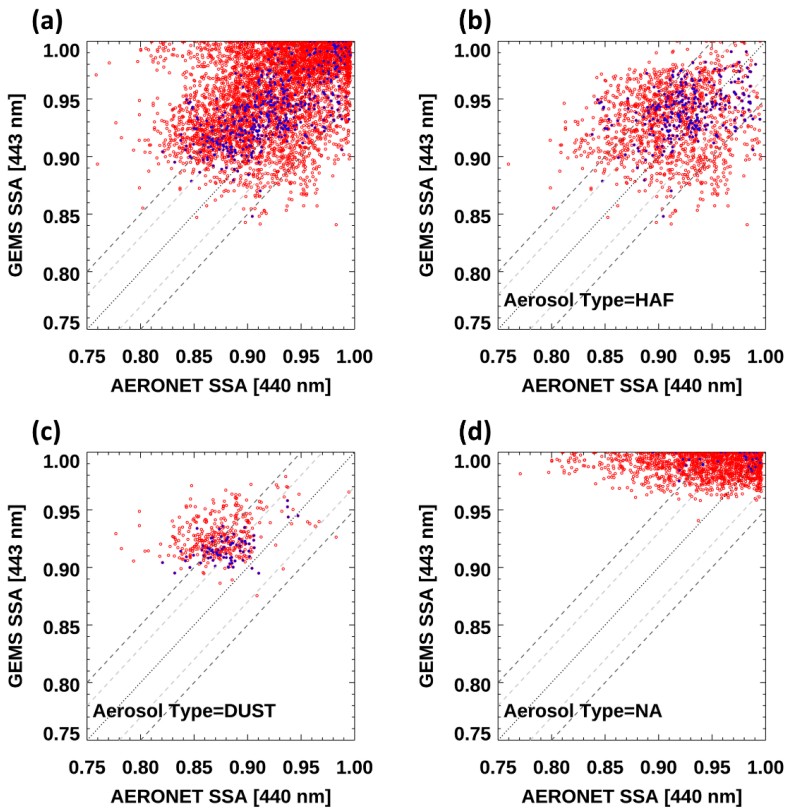

**Figure 9: Comparison of GEMS and AERONET SSA for (a) total and individual aerosol types: (b) HAF, (c) dust, and (d) NA. The red circles represent the pixels when AOD > 0.4, and the blue circles represent the pixels when AOD > 1.0. The gray dashed lines indicate an uncertainty envelope of ±0.03 in SSA, the black dashed lines indicate an uncertainty envelope of ±0.05 in SSA, and the dotted lines represent the 1:1 line. Data from November 1, 2021 to October 31, 2022 are used for comparison.**

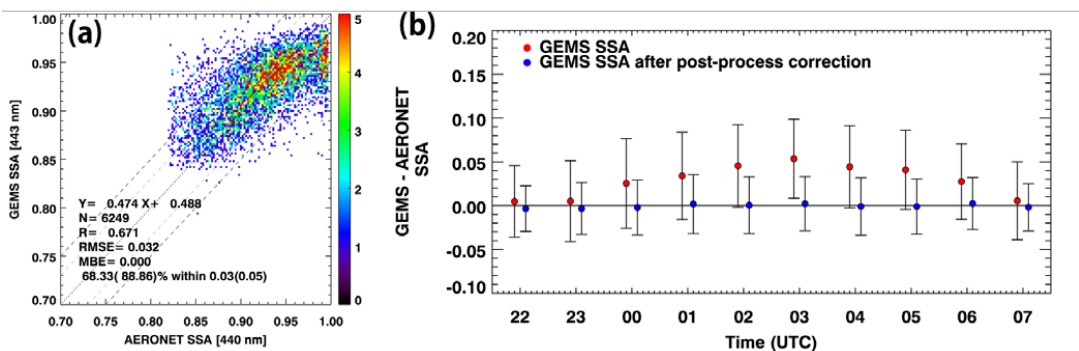

**Figure 10: (a) Comparison of GEMS SSA after post-process correction and AERONET SSA. The gray dashed lines indicate an uncertainty envelope of ±0.03 in SSA, the black dashed lines indicate an uncertainty envelope of ±0.05 in SSA, and the dotted lines represent the 1:1 line. (b) The difference between GEMS and AERONET SSA in terms of time. Data from December 1, 2021 to October 31, 2022 are used for comparison.**





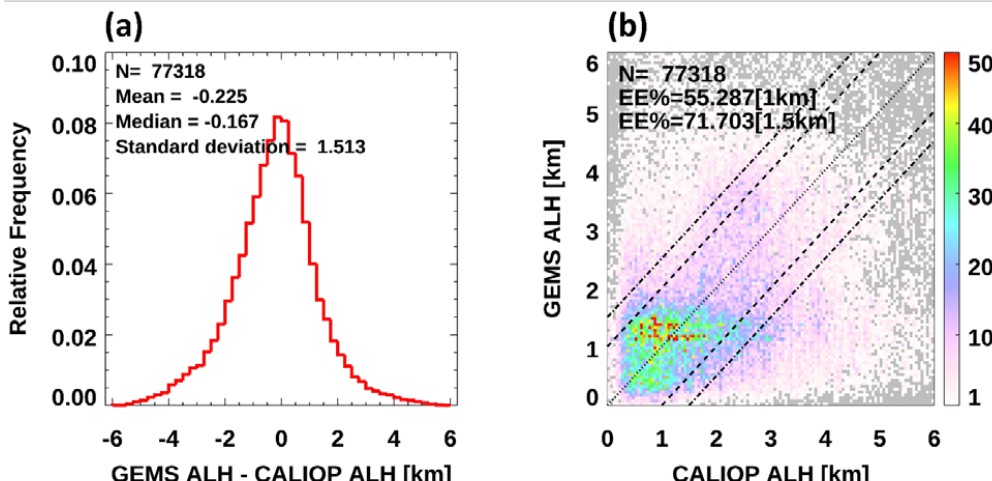


Figure 11: (a) Histogram of difference between GEMS and CALIOP ALH and (b) comparison of GEMS and CALIOP ALH. The
dashed lines indicate an uncertainty envelope of ±1 km in ALH. The dash-dotted lines indicate an uncertainty envelope of ±1.5 km
in ALH. The dotted lines represent the 1:1 line. Data from November 1, 2021 to October 31, 2022 are used for comparison.


Table 1:  Dimension of LUT in GEMS Aerosol algorithm.

| Variable Name [Unit] | Number of Entries | Entries |
|---|---|---|
| Wavelength [nm] | 6 | 354, 388, 412, 443, 477, 490 |
| SZA [°] | 12 | 0.01, 5, 10, 15, 20, 27, 34, 41, 48, 55, 62, 69 |
| VZA [°] | 12 | 0.01, 5, 10, 15, 20, 27, 34, 41, 48, 55, 62, 69 |
| RAA [°] | 11 | 0.01, 15, 30, 45, 60, 80, 100, 120, 140, 160, 180 |
| Surface reflectance [-] | 4 | 0.0, 0.05, 0.1, 0.2 |
| AOD at 443 nm [-] | 8 | 0.0, 0.1, 0.4, 0.8, 1.5, 2.0, 2.8, 3.6, 5.0, 10.0 |
| SSA at 443 nm [-] | 8 | 1.0, 0.98, 0.96, 0.94, 0.91, 0.88, 0.85, 0.82 for HAF and Dust 1.0, 0.99, 0.98, 0.97, 0.96, 0.94, 0.92, 0.90 for NA |
| ALH above the surface [km] | 5 | 0.5, 1.5, 3.0, 4.5, 6.0 |
| Elevation [km] | 3 | 0, 3, 6 |


Table 2: Statistic of hourly comparison of GEMS and AERONET AOD in Figure S2.

| Time | N | Slope | y-intercept | R | RMSE | MBE | Q (%) | GCOS (%) |
|---|---|---|---|---|---|---|---|---|
| 22:45 | 925 | 0.730 | 0.180 | 0.715 | 0.188 | 0.100 | 58.38 | 23.24 |
| 23:45 | 1964 | 0.684 | 0.190 | 0.830 | 0.212 | 0.076 | 59.32 | 20.93 |
| 00:45 | 4593 | 0.584 | 0.217 | 0.767 | 0.224 | 0.088 | 51.32 | 16.74 |
| 01:45 | 5632 | 0.534 | 0.200 | 0.774 | 0.211 | 0.054 | 54.83 | 17.47 |
| 02:45 | 6400 | 0.555 | 0.183 | 0.795 | 0.221 | 0.029 | 54.53 | 18.55 |
| 03:45 | 6139 | 0.569 | 0.165 | 0.824 | 0.233 | -0.013 | 56.54 | 17.04 |
| 04:45 | 6157 | 0.593 | 0.169 | 0.822 | 0.230 | 0.000 | 55.19 | 18.16 |
| 05:45 | 5642 | 0.586 | 0.204 | 0.773 | 0.235 | 0.041 | 52.87 | 19.25 |
| 06:45 | 4261 | 0.647 | 0.218 | 0.794 | 0.233 | 0.065 | 54.89 | 19.46 |
| 07:45 | 2541 | 0.617 | 0.224 | 0.771 | 0.247 | 0.054 | 56.55 | 19.48 |






Atmospheric
Measurement
Techniques



Discussions

**Table 3: Comparison of GEMS and AERONET SSA for different aerosol types in Figure 9. N represents the number of data, and**
**EE% denotes the percentage within the expected error range of ±0.03 (±0.05).**

| | GEMS AOD > 0.4 | | GEMS AOD > 1.0 | |
|---|---|---|---|---|
| Aerosol Type | N | EE% ±0.03 (±0.05) | N | EE% ±0.03 (±0.05) |
| All | 5227 | 42.76(67.25) | 454 | 56.61(83.70) |
| HAF | 1559 | 41.95(70.24) | 277 | 61.01(87.73) |
| Dust | 437 | 20.37(50.57) | 82 | 39.02(73.17) |
| NA | 1850 | 45.14(65.62) | 31 | 51.61(70.97) |


**Table 4: Statistic of comparison of GEMS and AERONET SSA in Figure S4.**

| | GEMS AOD > 0.4 | | GEMS AOD > 1.0 | |
|---|---|---|---|---|
| Time | N | EE% ±0.03 (±0.05) | N | EE% ±0.03 (±0.05) |
| 22:45 | 137 | 64.96 (86.13) | 23 | 52.17 (86.96) |
| 23:45 | 288 | 60.76 (83.68) | 67 | 74.63 (92.54) |
| 00:45 | 420 | 57.62 (82.38) | 93 | 73.12 (88.17) |
| 01:45 | 454 | 56.61 (79.07) | 113 | 63.72 (88.50) |
| 02:45 | 655 | 39.69 (62.90) | 237 | 45.99 (73.00) |
| 03:45 | 859 | 27.82 (53.20) | 339 | 25.07 (57.23) |
| 04:45 | 822 | 28.22 (55.60) | 335 | 27.76 (62.39) |
| 05:45 | 621 | 36.88 (63.12) | 222 | 38.29 (67.57) |
| 06:45 | 620 | 48.23 (73.23) | 255 | 51.37 (77.65) |
| 07:45 | 351 | 60.68 (79.49) | 160 | 63.12 (84.38) |
