# Peer review of "First Atmospheric Aerosol Monitoring Results from Geostationary"

_Atmospheric Measurement Techniques, 2023_

## Referee Comment (RC2)

[revised manuscript text omitted]

---

## Author Comment (AC1)

We appreciate the reviewer's insights and helpful comments, which improved the scientific quality of our manuscript. We carefully revised our manuscript basically reflecting reviewers' comments as much as we can. Our responses to the reviewer's comments are continued below with blue highlight. Please find our responses attached below.

Authors' response to RC1

Using observations from GEMS, the authors presented methods for retrieving aerosol optical properties, including AOD, SSA, ALH, UVAI and VisAI. Retrieved AOD, SSA and ALH data were evaluated against AERONET and CALIOP data. The concepts included in the study, including aerosol retrievals from UV and VIS observations, as well as using a machine learning method for noisy data removal, are not new. Still, this paper has some merits by applying the above mentioned methods to GEMS data. Still, there are major issues in this study that need to be addressed.

1. The post-processing step involves refining/correcting retrieved AOD and SSA values using AERONET data (1-30 days data before a given date) and through a machine leaning method (RF). This creates a potential issue, as non-trivial autocorrelations may exist in AERONET AOD and SSA data for a given AERONET station. Thus, by use the same AERONET site for training and testing, the results of the study may be biased toward AERONET sites. It is unknown the performance of the retrieved data over regions without AERONET data. I would suggest the authors pick some AERONET sites as the testing sites, and AERONET data from these test sites shall not be used for training purposes for the machine learning method.

We appreciate the reviewer's comment. We evaluated the post-processing results by separating the training and prediction periods temporally in the AERONET data. However, there could be spatial autocorrelation for AOD and SSA at the given AERONET station. To investigate the performance in areas without AERONET data, we conducted Leave-One-Site-Out Cross-Validation. The principle involves excluding the data from one site and training the model using the data from all other sites. The performance of the model is then evaluated using the data from the excluded site. The station chosen for evaluation is excluded from the model fitting process. For the period ranging from 30 days prior to the current day up to 1 hour before the target day, modeling is conducted, excluding data from site '*x*'. The model's predictive accuracy is then evaluated specifically for site '*x*' at the target day. Figure S2 shows the statistic maps illustrating the results of Leave-One-Site-Out Cross-Validation for post-process corrected GEMS AOD for the 1-year period of November 1, 2021, to October 31, 2022. In Northeast Asia, there is a notably high R, indicating a strong relationship in the AERONET data. However, sites closer to the equator tend to exhibit lower R values, around 0.5. The RMSE follows a similar pattern, with lower values in densely populated Northeast Asia, reflecting a better fit between predicted and AERONET values in this region. The MBE in Northeast Asia tends to be close to zero, suggesting minimal bias in the predictions. In contrast, the Indian region shows negative MBE values, indicating underestimation, while Southeast Asia has positive values, signifying overestimation. Therefore, our post-processing method may have the potential to have decreased accuracy in areas without (or with few) AERONET sites.

[Figure]

**Figure S2: The statistic maps illustrating the results of site-based cross-validation for post-process corrected GEMS AOD for the 1-year period of November 1, 2021, to October 31, 2022.**

2. Also, cloud contamination exists prior to the post-processing step (Figure 7) yet is suppressed by the post processing step (Figure 8). Any reason for that? Could this be potentially causing an issue? Those cloud contaminated pixels should be excluded in the study.

Thanks for careful comments. As the reviewer pointed out, we checked that pixels contaminated by clouds were included during the post-processing step. Including cloud-contaminated pixels in the modelling of the post-processing step could lead to improper training and should therefore be excluded. We have revised our spatiotemporal collocation criteria to exclude such cloud-contaminated pixels during the modelling process, considering better cloud masking of AERONET. When an AERONET site is located within a GEMS pixel, AERONET data are temporally matched within a ±10-minute window of the GEMS observation time. Data from three AERONET sites (Sorong, Jambi and BMKG_GAW_PALU) with severe sub-pixel cloud contamination were excluded from the training. By strictly applying criteria to the data used for removing cloud-contaminated pixels from the modelling process, we achieved an enhancement in prediction performance. The R value increases from 0.899 to 0.920, and the Q value also raised from 79.13% to 82.17%.

[Figure]

**Figure 8: (a) Comparison of GEMS AOD after post-process correction by machine learning and AERONET AOD. The dashed lines indicate an uncertainty envelope of a larger 0.1 or ±30% in AOD. The dotted lines represent the 1:1 line. The difference between GEMS AOD and AERONET AOD in terms of time. (b) All pixels, (c) pixels when AERONET AOD < 0.4, and (d) pixels when AERONET AOD > 0.4. The red circles represent the GEMS AOD, and the blue circles represent the GEMS AOD after post-process correction. The error bars correspond to the standard deviation. Data from November 1, 2021 to October 31, 2022 are used for comparison.**

3. Version 3, level 1.5 AERONET data were used in this study. I would recommend the authors use version 3, level 2 AERONET data as it is quality assured. There is a reason why the AERONET team spent efforts creating level 2 data from level 1.5 data. The additional data included in the level 1.5 AERONET data may likely be problematic retrievals.

We appreciate the reviewer's helpful comments. We agree that AERONET Level 2.0 data ensures higher quality compared to Level 1.5. With your suggestion, we have updated the validation results using Level 2.0 data (Figure7, Figure 9, Table 2, Table 3 and Table 4). However, using AERONET Level 2.0 data in the post-processing step presents two major challenges: 1) There is a significant reduction in the volume of data available for the modelling process, with a particularly noticeable impact on SSA. The amount of data has decreased to such an extent that it affects the modelling and prediction processes (a reduction to ~1/4th of the original volume). 2) Near-real-time modelling has become impossible. Given the focus on post-processing process in near-real-time, the second issue in particular poses risks. Due to these concerns, we decided to continue using AERONET level 1.5 data for post-processing step (Figure 8 and Figure 10). Additionally, the data previously used for comparison from December 1, 2021, to October 31, 2022 (11 months). To extend the validation period to one year, we have updated the comparison to include data from November 1, 2021, to October 31, 2022.

[revised manuscript text omitted]

4. The paper needs to be carefully proof-read. There are quite a few places that I tried to guess what the authors were trying to say.

Done. Thank you.

5. It would be interesting to show seasonal and regional variations of AOD, SSA and ALH as a function of local time (diurnal patterns). It shall not be a difficult task as the authors have the data handy for the task.

Thanks for careful comments. Figure S7 shows seasonal and regional variation as a function of UTC for each of the following four regions: Korea (33° N–39° N and 124° E–132° E), North China (33° N–34° N and 110° E–124° E), South China (21° N–33° N and 110° E–122° E), Indochina peninsula (8° N–22° N and 92° E–110° E). The Indian region was excluded from the regional analysis because the observable area within the total region of India varies significantly depending on the GEMS scan profiles. After gridding the GEMS AOPs into a 0.1° × 0.1° grid box, monthly averages were calculated. After the monthly averaging, seasonal averages were calculated for each pixel, but only where all three months within a season had data available for the given pixel. Regional averages were conducted only when more than 50% of the available values within the domain. For AOD, a U-shaped or flat diurnal variation was observed in all four regions. In the case of SSA, higher values were observed during JJA (June, July, August) in Korea, North China, and South China, which is considered to be influenced by aerosol hygroscopic growth due to relatively high atmospheric humidity. However, the Indochina Peninsula showed the highest SSA values in SON (September, October, November) and the lowest in DJF (December, January, February), which is consistent with the relatively low SSA values observed at the Chiang Mai AERONET site from 2011 to 2016 during DJF (Liang et al., 2019). However, there are limitations in investigation of diurnal variation for ALH. The diurnal variations of ALH were not consistent with those of mixing layer height. One reason for the uncertainty in ALH is that it is retrieved from OE, depending on the uncertainty of *a priori* AOD, SSA and ALH. Before post-processing, GEMS AOD and SSA exhibited diurnal biases pattern compared to AERONET data (details in Sections 5.1 and 5.2). Those uncertainties cloud affect the uncertainty in the diurnal variation of ALH. Since GEMS ALH cannot be post-corrected using CALIOP data (details in Section 3), we are considering post-process corrected ALH using ground-based lidar observation networks (i.e., Korea Aerosol Lidar Observation Network; the Asian dust and aerosol lidar observation network) in the future study. Therefore, one of the limitations of the paper is that GEMS ALH has limitations in the detailed investigation diurnal variations of ALH due to its much weaker signals.

[Figure]

**Figure S7: Seasonal and regional AOD, SSA, and ALH variation as a function of UTC for each of the following four regions: Korea (33° N–39° N and 124° E–132° E), North China (33° N–34° N and 110° E–124° E), South China (21° N–33° N and 110° E–122° E), Indochina peninsula (8° N–22° N and 92° E–110° E) during the period of November 1, 2021, to October 31, 2022. The yellow lines represent spring MAM: March, April, and May), and the blue lines represent summer (JJA: June, July, and August), and the red lines represent autumn (SON: September, October, and November), and the green lines represent and winter (DJF: December, January, and February).**

Minor comments are corrected.

Page 1, lines 30 and 34. "GEMS AOD". Wavelength for the mentioned AOD?

Done. Thank you.

Page 2, line 46, "While significant diurnal variations in AOPs have been observed". Provide references.

Done. Thank you.

Page 3, line 82,"Considering the solar zenith angle". What do the authors mean? Considering the sun position changes??

I have clarified the sentence for better understanding. Thank you.

Page 3, lines 96-97, I am not sure what the authors try to express. Please try to rewrite.

I have clarified the sentence for better understanding. Thank you.

Page 3, line 115, "In this paper, we report the first aerosol monitoring results"?? Aerosol retrievals?? Please rewrite.

I have clarified the sentence for better understanding. Thank you.

Page 4, lines 125-126 make no sense to me. Please try to rewrite.

I have clarified the sentence for better understanding. Thank you.

Page 4, equation 1. Where is SZA? Downward solar radiation shall be a function of SZA.

The term "normalized radiance" mentioned in this paper refers to the "sun-normalized radiance" as defined in the OMI Algorithm Team's Terms and Symbols (https://eospso.nasa.gov/sites/default/files/atbd/ATBD-OMI-Terms-Symbols.pdf). Therefore, the Solar Zenith Angle (SZA) is not required.

Page 5, line 166, "The preliminary GEMS AERAOD".. Shall be "An early version of GEMS AERAOD"??

Done. Thank you.

Page 5, line 190.   Define "the Levenberg-Marquardt equation". Or provide a reference.

Done. Thank you.

Page 6, line 208, "the calculations were performed using the Mie theory"   Be specific.   I assume the authors computed optical properties using the Mie code and applied the computed optical properties in RTM calculations. What about needed parameters for Mie and RTM simulations? But please be precise with your discussions.

Done. Thank you.

Page 6, line 211, provide references for the GEMS spectral response function.   Also, GEMS has a spectral resolution of 0.6 nm. Why do the authors resample the spectral data (from RTM) to a spectral resolution of 0.2 nm?

GEMS measures radiance/irradiance in ultraviolet and visible wavelengths with 0.2 nm spectral sampling and about 0.6 nm FWHM resolution. Therefore, we resampled according to 0.2 nm spectral sampling.

Page 6, line 226, "minimum reflectance method" Need a reference here.

Done. Thank you.

Page 7, line 242, this equation doesn't make sense. Please check.

Done. Thank you.

Page 7, Section 2.1.4, may need plots to demonstrate cloud detection steps.

I have clarified the sentence for better understanding. Thank you.

Page 10, line 379.   Define Q value.   Be specific about GCOS requirement.

Done. Thank you. (Page8 Line 283-286.)

---

## Author Comment (AC2)

We appreciate the reviewer's insights and helpful comments, which improved the scientific quality of our manuscript. We carefully revised our manuscript basically reflecting reviewers' comments as much as we can. Our responses to the reviewer's comments are continued below with blue highlight. Please find our responses attached below.

Authors' response to RC2

Reviewer #2: In this paper, authors presented the first results of a suite of operational aerosol products, including AOD, SSA and ALH, from Geostationary Environment Monitoring Spectrometer (GEMS), through evaluating their performance in monitoring air pollution events over Asia, and through validation against AERONET and CALIOP data. The objective of this paper is very clear, which intends to show the performance of operational GEMS aerosol product under different scenarios. The methodology in this paper is sound, solid and moderately innovative. The presentation of this paper is relatively clear, but could be largely improved. The results are significant enough to be published after some revisions. However, the manuscript needs to be reorganized to increase the clarity, some missing/misplaced figures needs to be fixes, also contains many editorial errors. It is suggested to be accepted but after some revisions.

1. The core of this paper, in my opinion, is to demonstrate the performance of the GEMS operational aerosol retrieval algorithm, which went through some updates on the original version. The updates include new hourly surface reflectance database derived from GEMS observations by using the minimum reflectance in 30 days and the background AOD derived from AERONET measurements, new cloud screening method and postprocessing approach using machine learning technique to reduce not only biases but also time-dependency of the biases (As shown in Figure 1). Therefore, descriptions of those updates (bold box in Figure 1) and GEMS algorithm itself deserve to be in more details. The authors have described/touched all aspects of the updates, but is slightly lack of details. The questions should be answered in more details include how these updates are designed and applied, and most importantly what are the impacts on the retrievals. Some of the contents in section 5 can be moved here and show the impact/improvement after algorithm updates. For example, section 2.13, 2.14 and 3, all should add examples to show the improvement. And, It is suggested that section 4 and 5 will solely focus on the evaluations of results from the latest version of algorithm.

Thanks for your comments. Based on minor comments, we further detailed the description of the GEMS aerosol algorithm in the paper. Some of the contents in section 5 were moved. In addition, we analyze how each update factor has influenced the AOD validation results. The validation period is January, April, and July of 2022. The early version of GEMS AERAOD exhibited an R of 0.511 and a Q-value of 16.27%. When using KNMI irradiance instead of GEMS irradiance and changing to spectral binning LUTs, Set1 resulted in a closer MBE of -0.074 to zero and an increased Q-value of 50.63%, approximately 30% higher than results of the early version of GEMS AERAOD. Set2, using GEMS surface reflectance, showed a slight decrease in the R-value but an improvement in the Q-value by over 7%. Finally, introducing a new cloud removal method (Set3) increased the R and decreased the RMSE, leading to an increase in the Q-value compared to Set2.

**Table S1: Statistics of comparison of GEMS and AERONET AOD at 443 nm. The validation period is January, April, and July 2022. Set1 refers to the application of Section 2.1.2 (spectral binning and KNMI irradiance) to the early version of GEMS AERAOD. Set2 means applying Section 2.1.3 (GEMS surface reflectance) to Set1. Set3 implies the application of Section 2.1.4 (new cloud masking) to Set2.**

|  | The early version of GEMS AERAOD | Set1 | Set2 | Set3 |
|---|---|---|---|---|
| N | 11100 | 12321 | 10065 | 9874 |
| Slope | 0.462 | 0.735 | 0.656 | 0.664 |
| y-intercept | 0.557 | 0.034 | 0.103 | 0.095 |
| R | 0.511 | 0.754 | 0.740 | 0.768 |
| RMSE | 0466 | 0.274 | 0.262 | 0.249 |
| MBE | 0.369 | -0.074 | -0.037 | -0.044 |
| Q (%) | 16.27 | 50.63 | 57.22 | 57.88 |
| GCOS (%) | 5.61 | 16.65 | 19.81 | 20.16 |

2. Quality of the Figures needs to be improved, especially Figure 2, Figure 5 and 6. In addition, Figure S1, S2, S3 and S4 are described in text and the actually figures are missing.

Thank you for your feedback on the figures in our paper. We acknowledge the concerns regarding the quality of Figures 2, 5, and 6 and missing figures. We enhanced the clarity and resolution of Figures 2, 4, 5, and 6 and we included the missing supplemental figures (Figure S1, S2, S3, and S4). During the revision, many supplementary materials were added. Figure 2 originally displayed AOD, SSA, ALH, UVAI, and VisAI over time, but due to issues with distinguishability in the graphic, it has been modified to only show AOD, SSA, and ALH. Figure 4 and Figure 5 are enlarged for improved image quality and visibility. Figure 6 is updated to be AERONET version 3 Level 2.0, and the site symbols are significantly enlarged.

[Figure]

**Figure 2: Hourly GEMS aerosol products for the dust case on April 8, 2022 over northwestern China. Time-series maps of AOD at 443 nm, SSA at 443 nm and ALH from 22:45 to 07:45. The circle denotes an AERONET station, and the filled color indicates the AERONET AOD and SSA at 443 nm in the AOD and SSA columns. GEMS SSA, and ALH are displayed only when GEMS AOD > 0.2.**

[Figure]

**Figure 2: Continue**

[Figure]

**Figure 2: Continue**

[Figure]

**Figure 4: The example of the GEMS AOD before and after post-processing for an absorbing aerosol case over Indo-Gangetic Plane at 04:45 UTC on December 4, 2021. (a) GEMS false RGB. The circle denotes an AERONET station, and the filled color indicates the AERONET AOD at 443 nm, (b) GEMS AOD and (c) GEMS AOD after post-process correction.**

[Figure]

**Figure 5: The example of GEMS SSA and the GEMS SSA after post-processing for an absorbing aerosol case over India, Bangladesh, and mainland Southeast Asia at 03:45 UTC on December 23, 2021. (a) GEMS false RGB. The circle denotes an AERONET station, and the filled color indicates the AERONET SSA at 440 nm, (b) GEMS SSA, and (c) GEMS SSA after post-process correction.**

[Figure]

AERONET sites used for the GEMS AOD and SSA validation

01:Dalanzadgad( 502, 227)
02:Hokkaido_University( 345, 128)
03:Beijing-CAMS( 291, 137)
04:XiangHe( 334, 0)
05:Baengnyeong( 427, 0)
06:Niigata( 342, 116)
07:Gangneung_WNU(1181, 353)
08:Yonsei_University(1053, 399)
09:Seoul_SNU(1035, 0)
10:Hankuk_UFS( 997, 214)
11:Noto( 299, 0)
12:DRAGON_Hakuba( 5, 0)
13:Anmyon( 763, 0)
14:DRAGON_Omachi( 7, 0)
15:DRAGON_Mt_Haruna( 2, 0)
16:DRAGON_Takayama( 31, 2)
17:DRAGON_Matsumoto( 440, 203)
18:TGF_Tsukuba( 270, 142)
19:DRAGON_Suwa( 10, 4)
20:DRAGON_Minowa( 74, 32)
21:DRAGON_Ina( 71, 42)
22:DRAGON_Kofu( 312, 78)
23:Chiba_University( 470, 152)
24:KORUS_UNIST_Ulsan(1125, 612)
25:DRAGON_Iida( 72, 29)
26:Gwangju_GIST( 428, 0)
27:Osaka( 383, 0)
28:Fukuoka( 998, 375)
29:Gosan_NIMS_SNU( 148, 69)
30:Fukue( 240, 38)
31:Lahore( 842, 409)
32:IIT_Delhi( 101, 90)
33:QOMS_CAS( 327, 0)

34:Amity_Univ_Gurgaon( 607, 0)
35:Pokhara( 333, 119)
36:Dibrugarh_Univ( 472, 214)
37:Jaipur( 54, 0)
38:Okinawa_Hedo( 874, 133)
39:Kanpur( 114, 86)
40:Gandhi_College( 96, 78)
41:Cape_Fuguei_Station( 53, 33)
42:Taipei_CWB( 433, 0)
43:EPA-NCU( 267, 121)
44:TASA_Taiwan( 680, 157)
45:Xitun( 579, 0)
46:Dhaka_University( 651, 220)
47:Lulin( 575, 16)
48:Chen-Kung_Univ( 792, 0)
49:Kaohsiung( 957, 0)
50:Hong_Kong_Sheung( 30, 0)
51:Hong_Kong_PolyU( 603, 0)
52:Bhola( 485, 256)
53:Dongsha_Island( 219, 28)
54:Doi_Ang_Khang( 147, 40)
55:Nong_Khai( 603, 104)
56:Bangkok( 256, 0)
57:Sra_Kaeo( 394, 0)
58:Chachoengsao( 239, 0)
59:Tai_Ping( 43, 1)
60:USM_Penang( 347, 30)
61:Singapore( 413, 8)
62:Pontianak( 39, 0)
63:Sorong( 163, 0)
64:Jambi( 222, 0)
65:BMKG_GAW_PALU( 134, 5)
66:Makassar( 19, 0)

**Figure 6: AERONET sites used for the GEMS AOD and SSA validation. The red color indicates the site where validation points exist for both AOD and SSA. The green color indicates the site where validation points exist only for AOD. The list of station names in conjunction with the number of AERONET AOD and SSA data points for validation at each station.**

[Figure]

**Figure S1: Monthly BAOD at 443 nm from 2-year AERONET AOD and interpolated to a 0.1 × 0.1° box. The lowest fifth percentiles of the AERONET AOD 443 nm values at each AERONET site are plotted as circles for comparison.**

[Figure]

[Figure]

[Figure]

**Figure S2: The statistic maps illustrating the results of site-based cross-validation for post-process corrected GEMS AOD for the 1-year period of November 1, 2021, to October 31, 2022.**

[Figure]

**Figure S3: Average variable ranking from RF model for the post-processing correction of GEMS AOD at 443 nm for the 1-year period of November 1, 2021, to October 31, 2022.**

.

[Figure]

**Figure S4: Average variable ranking from RF model for the post-processing correction of GEMS SSA at 443 nm for the 1-year period of November 1, 2021, to October 31, 2022.**

.

[Figure]

Figure S5: Hourly comparison of GEMS and AERONET AOD at 443 nm for the 1-year period of November 1, 2021, to October 31, 2022. The dashed lines indicate an uncertainty envelope of maximum (0.1 or 30%) in AOD. The dotted lines represent the 1:1 line.

[Figure]

**Figure S6: Hourly comparison of GEMS and AERONET SSA for the 1-year period of November 1, 2021, to October 31, 2022. The red circles represent the pixels when AOD > 0.4, and the blue circles represent the pixels when AOD > 1.0. The gray dashed lines indicate an uncertainty envelope of ±0.03 in SSA, the black dashed lines indicate an uncertainty envelope of ±0.05 in SSA, and the dotted lines represent the 1:1 line.**

[Figure]

**Figure S7: Seasonal and regional AOD, SSA, and ALH variation as a function of UTC for each of the following four**

**regions: Korea (33° N–39° N and 124° E–132° E), North China (33° N–34° N and 110° E–124° E), South China (21° N–33° N and 110° E–122° E), Indochina peninsula (8° N–22° N and 92° E–110° E) during the period of November 1, 2021, to October 31, 2022. The yellow lines represent spring MAM: March, April, and May), and the blue lines represent summer (JJA: June, July, and August), and the red lines represent autumn (SON: September, October, and November), and the green lines represent and winter (DJF: December, January, and February).**

**Table S1: Statistics of comparison of GEMS and AERONET AOD at 443 nm. The validation period is January, April, and July 2022. Set1 refers to the application of Section 2.1.2 to the early version of GEMS AERAOD. Set2 means applying Section 2.1.3 to Set1. Set3 implies the application of Section 2.1.4 to Set2.**

|  | The early version of GEMS AERAOD | Set1 | Set2 | Set3 |
|---|---|---|---|---|
| N | 11100 | 12321 | 10065 | 9874 |
| Slope | 0.462 | 0.735 | 0.656 | 0.664 |
| y-intercept | 0.557 | 0.034 | 0.103 | 0.095 |
| R | 0.511 | 0.754 | 0.740 | 0.768 |
| RMSE | 0466 | 0.274 | 0.262 | 0.249 |
| MBE | 0.369 | -0.074 | -0.037 | -0.044 |
| Q (%) | 16.27 | 50.63 | 57.22 | 57.88 |
| GCOS (%) | 5.61 | 16.65 | 19.81 | 20.16 |

3. It is strongly suggested to go through editorial revision, including language, terms and sentence structures etc.

Thanks for your comments. Done. Thank you.

Minor comments are corrected.

Line 48 1 Per day    or    daily may be more accurate?

Done. Thank you.

line 61 This statement is a bit confusing. Rayleigh Scattering itself can not be reduced due to aerosols. It is the contribution of R1ayleigh scattering to the satellite observations.

Done. Thank you. "The contribution of Rayleigh scattering to the total Top of the Atmosphere (TOA) reflectance enhancements is reduced below the aerosol layer owing to aerosol attenuation (Kayetha et al., 2022; Torres et al., 2005)."

Line 89 "Before GEMS launches?", add "first" ?

Done. Thank you.

Line 93-96 This part needs to be rephrased to indicate that the better GEMS AOD retrieval is achieved by taking into account the spectral dependence of aerosol absorption in the UV-Vis region, which was considered as "independent" of wavelength in the previous version...

Done. Thank you. "Spectral variations of aerosol absorption in the UV-Vis region, as investigated by Go et al. (2020a), are applied to the GEMS aerosol algorithm to achieve improved AOPs retrieval. This adjustment accounts for the spectral dependence of aerosol absorption, previously treated as independent of wavelength."

Line 96-97 please add one or two sentence to describe    what the findings are ...

Done. Thank you. "To improve the accuracy of GEMS aerosol retrieval, Go et. al. (2020b) tested the use of cloud mask information from MODIS IR channels for removing cirrus and sub-pixel cloud contamination, as well as the total dust confidence index for classification of aerosol type. The limitations associated with the UV-Vis regions of GEMS were overcome by using the IR channels of other satellites, leading to the application of research on synergistic use of hyperspectral satellite instrument and broadband meteorological imager.

Line 100 "is shown as"?     please rephrase tis sentence....

Done. Thank you.

Line 104-106 suggestion:

Zhang et. al. (2020)    developed an empirical AOD bias-correction algorithm, which utilizes the lowest AOD.............

Done. Thank you.

Line 115 from GEMS operational observations ?

Done. Thank you.

Line 189 "with an ALH based on the climatology of CALIOP ALH" ???

Done. Thank you.

Line 198 "the same "

Done. Thank you.

Line 199 "are different from"? please rephrase this sentence.

Done. Thank you.

Line 205 owing to

Done. Thank you.

Line 213 what range?

Done. Thank you.

Line 220-223 These three sentences can be combined into one sentence and will be much easier for reader to read. Example is given as following:

"The preliminary GEMS AERAOD retrieval algorithm used the OMI surface reflectance climatology data product (OMLER v003) (Kleipool et al. 2008), with a spatial resolution of $0.5 \times 0.5°$,    which is too coarse compared with GEMS pixel size, therefore, resulting in discontinuities in the GEMS AOPs.    "

Done. Thank you.

Line 233: This section is not described clearly. How the BAOD is considered in the retrieval algorithm? is it used to derived surface reflectance dataset? if so, please described in details..

Done. Thank you.

"Rayleigh, gaseous absorption, and BAOD are corrected in the atmospheric correction process to create a surface reflectance dataset."

Line 235: retrievals???

Done. Thank you.

Line 247: missing?

Done. Thank you.

Line 259: which band???

Done. Thank you.

Line 262: Can you explain it in more details? what the contrast mean? how?

Done. Thank you.

Line 265-267: This part is really confusing, please reorganize these sentences and give in more clear descriptions on the various steps.....

such as " (5-2) after (3-1)", what does this mean?

Done. Thank you.

Line 282: "which are calculated"

Done. Thank you.

Line 296: " limited by the fact that"

Done. Thank you.

Line 304: how about the other parameters?

Done. Thank you.

Line 307: Do you mean, "the GEMS retrieval domain coverage changes with the time due to the varying GEMS SCAN patterns with the SZA?"

Done. Thank you.

Line 310: "reaching to 2.0"?

Done. Thank you.

Line 331: higher VisAI indicated coarse particle, is it true for biomass burning event? please explain it in more details

Done. Thank you.

"GEMS VisAI didn't clearly show signals from small particle sizes caused by biomass burning, indicating that signals from the surface were not completely removed. There may be limitations in considering aerosol size information using GEMS VisAI (Go et al., 2020b)."

Line 353: It is suggested to rephrase    this sentence. "Post-processed AOD shown an elevated value, especially in the moderate original AOD range (~0.7), bring the GEMS AOD closer to AERONET AOD"

Done. Thank you.

Line 358: rephrase this sentence as " The GEMS false color RGB with AERONET stations represented by circles is given in Figure 5a"

Done. Thank you.

Line 405: Can not find Figure S2 in paper...

Done. Thank you.

Line 441: missing?

Done. Thank you.

Line 476: missing?

Done. Thank you.

Line 765: UTC or local time? please indicate them here...

Done. Thank you.